# Existence and functions of a kisspeptin neuropeptide signaling system in a non-chordate deuterostome species

Tianming Wang[1,2†]*, Zheng Cao[3†], Zhangfei Shen[3†], Jingwen Yang[1,2], Xu Chen[1], Zhen Yang[1], Ke Xu[1], Xiaowei Xiang[1], Qiuhan Yu[1], Yimin Song[1], Weiwei Wang[3], Yanan Tian[3], Lina Sun[4], Libin Zhang[4,5], Su Guo[2], Naiming Zhou[3]*

[1]National Engineering Research Center of Marine Facilities Aquaculture, Marine Science College, Zhejiang Ocean University, Zhoushan, China; [2]Programs in Human Genetics and Biological Sciences, Department of Bioengineering and Therapeutic Sciences, University of California, San Francisco, San Francisco, United States; [3]Institute of Biochemistry, College of Life Sciences, Zijingang Campus, Zhejiang University, Hangzhou, China; [4]Key Laboratory of Marine Ecology and Environmental Sciences, Institute of Oceanology, Chinese Academy of Sciences, Qingdao, China; [5]Center for Ocean Mega-Science, Chinese Academy of Sciences, Qingdao, China

*For correspondence:
wangtianming@zjou.edu.cn (TW);
zhounaiming@zju.edu.cn (NZ)

†These authors contributed equally to this work

**Competing interests:** The authors declare that no competing interests exist.

**Abstract** The kisspeptin system is a central modulator of the hypothalamic-pituitary-gonadal axis in vertebrates. Its existence outside the vertebrate lineage remains largely unknown. Here, we report the identification and characterization of the kisspeptin system in the sea cucumber *Apostichopus japonicus.* The gene encoding the kisspeptin precursor generates two mature neuropeptides, AjKiss1a and AjKiss1b. The receptors for these neuropeptides, AjKissR1 and AjKissR2, are strongly activated by synthetic *A. japonicus* and vertebrate kisspeptins, triggering a rapid intracellular mobilization of $Ca^{2+}$, followed by receptor internalization. AjKissR1 and AjKissR2 share similar intracellular signaling pathways via $G_{\alpha q}$/PLC/PKC/MAPK cascade, when activated by C-terminal decapeptide. The *A. japonicus* kisspeptin system functions in multiple tissues that are closely related to seasonal reproduction and metabolism. Overall, our findings uncover for the first time the existence and function of the kisspeptin system in a non-chordate species and provide new evidence to support the ancient origin of intracellular signaling and physiological functions that are mediated by this molecular system.

## Introduction

Nervous systems, from simple nerve nets in primitive species to complex architectures in vertebrates, process sensory stimuli and enable animals to generate body-wide responses (*Arendt et al., 2016*). Neurosecretory centers, one of the major output systems in the animal brain, secrete neuropeptides and nonpeptidergic neuromodulators to regulate developmental, behavioral and physiological processes (*Tessmar-Raible, 2007*). Understanding the evolutionary origin of these centers is an area of active investigation, mostly because of their importance in a range of physical phenomena such as growth, metabolism, or reproduction (*Tessmar-Raible et al., 2007*; *Zandawala et al., 2017*).

The hypothalamus constitutes the major part of the ventral diencephalon in vertebrates and acts as a neurosecretory brain center, controlling the secretion of various neuropeptides (hypothalamic neuropeptides) (*Bakos et al., 2016*; *Burbridge et al., 2016*). Beyond vertebrates, similar neurosecretory systems have been seen in multiple protostomian species, including crustaceans, spiders,

and mollusks (*Hartenstein, 2006*). Specific to echinoderms, which occupy an intermediate phylogenetic position asdeuterostomian invertebrate species with respect to vertebrates and protostomes, increasing evidence, collected from in silico identification of hypothalamic neuropeptides and the functional characterization of a vasopressin/oxytocin (VP/OT)-type signaling system, suggests the existence of conserved signaling elements (*Zandawala et al., 2017*; *Odekunle et al., 2019*).

The hypothalamic neuropeptide kisspeptins, encoded by the *Kiss1* gene and most notably expressed in the hypothalamus, share a common Arg-Phe-amide motif at their C-termini and belong to the RFamide peptide family (*Roseweir and Millar, 2009*; *Uenoyama et al., 2016*). Exogenous administration of kisspeptins triggers an increase in the circulating levels of gonadotropin-releasing hormone and gonadotropin in humans, mice, and dogs (*Gottsch et al., 2004*; *Dhillo et al., 2005*; *Dhillo et al., 2007*; *Albers-Wolthers et al., 2014*). Accumulating evidence suggests that the kisspeptin system functions as a central modulator of the hypothalamic-pituitary-gonadal (HPG) axis to regulate mammalian puberty and reproduction through a specific receptor, GPR54 (also known as AXOR12 or hOT7T175), which is currently referred to as the kisspeptin receptor (*Muir et al., 2001*; *Kirby et al., 2010*; *Javed et al., 2015*). Following the discovery of kisspeptins and kisspeptin receptors in mammals, a number of kisspeptin-system paralogous genes have been revealed in other vertebrates (*Pasquier et al., 2014*), and a couple of functional kisspeptin and its receptor have also been demonstrated in amphioxus (*Wang et al., 2017*). Moreover, kisspeptin-type peptides and their corresponding receptors from echinoderms have been annotated in silico, on the basis of the analysis of genome and transcriptome sequence data (*Mirabeau and Joly, 2013*; *Elphick and Mirabeau, 2014*; *Semmens et al., 2016*; *Semmens and Elphick, 2017*; *Suwansa-Ard et al., 2018*; *Chen et al., 2019*). However, to our knowledge, neither the kisspeptin-type peptides nor the corresponding receptors have been experimentally identified and functionally characterized in non-chordate invertebrates. This raises an important question: does the kisspeptin signaling system have an ancient evolutionary origin or did it evolve de novo in the chordate/vertebrate lineages?

Here, we addressed this question by searching for kisspeptin and its receptor genes in a non-chordate species, the sea cucumber *Apostichopus japonicus*. It is one of the most studied echinoderms and is widely distributed in temperate habitats in the western North Pacific Ocean, being cultivated commercially on a large scale in China (*Purcell et al., 2012*). We uncovered kisspeptin-like and kisspeptin receptor-like genes by mining published *A. japonicus* data (*Zhang et al., 2017*; *Chen et al., 2019*) using a bioinformatics approach. Their signaling properties were characterized using an in vitro culture system. Through the evaluation of $Ca^{2+}$ mobilization and other intracellular signals, we found that *A. japonicus* kisspeptins activate two receptors (AjKissR1 and AjKissR2) via a GPCR-mediated $G_{\alpha q}$/PLC/PKC/MAPK signaling pathway in a mammalian cell line. Although likely, it remains to be determined whether the same signaling cascade also occurs in vivo in its seemingly conserved function in reproductive control. Finally, we revealed the physiological activities of this signaling system both in vivo and ex vivo, and we demonstrated the involvement of the kisspeptin system in reproductive and metabolic regulation in *A. japonicus*. Collectively, our findings indicate the existence of a kisspeptin signaling system in non-chordate deuterostome invertebrates and provide new evidence to support the ancient evolutionary origin of the intracellular signaling and physiological functions mediated by this kisspeptin system (*Tessmar-Raible et al., 2007*).

## Results

### In silico identification of kisspeptins and kisspeptin receptors

The putative *A. japonicus* kisspeptin precursor gene (*AjpreKiss*) was identified in silico from transcriptome data and cloned from anterior part (ANP, containing the nerve ring) tissue samples by reverse transcription polymerase chain reaction (RT-PCR). The full-length cDNA (GenBank accession number MH635262) was 2481-bp long and contained a 543-bp open reading frame (ORF), encoding a 180-amino-acid peptide precursor (AjpreKiss) with one predicted signal peptide region and four cleavage sites (*Figure 1A* and *Figure 1—figure supplement 1*). Two mature peptides with amide donors for C-terminal amidation—a 32-amino-acid kisspeptin-like peptide with a disulfide-bond (AjKiss1a) and an 18-amino-acid kisspeptin-like peptide (AjKiss1b)—were predicted (sequences listed in the 'Key Resources Table'). The organization of the *AjpreKiss* gene as determined from the genome, as well as those of the zebrafish *DrpreKiss1*, *DrpreKiss2* and human *HspreKiss1*, showed two exons in the

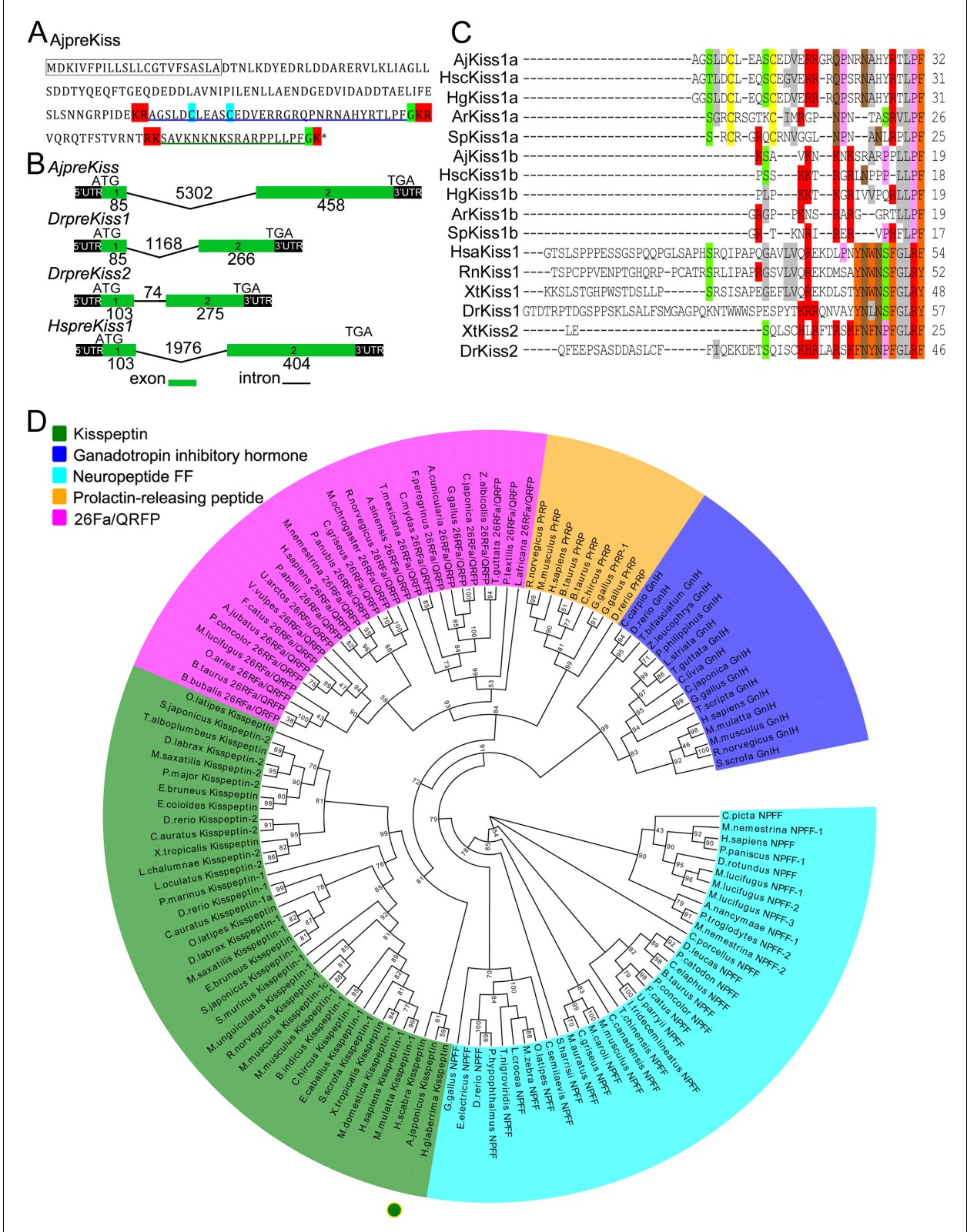

**Figure 1.** Gene structure, homology, and phylogenetic characterization of *Apostichopus japonicus* kisspeptin precursor (AjpreKiss). (**A**) Deduced amino-acid sequence of AjpreKiss. The signal peptide is labeled in the box with full lines; the cleavage sites are highlighted in red; glycine residues responsible for C-terminal amidation are highlighted in green; cysteines paired in a disulfide-bonding structure are highlighted in light blue; the predicted mature peptides, AjKiss1a and AjKiss1b, are noted by the blue and green underlines. (**B**) The organization of the *AjpreKiss* gene is compared

*Figure 1 continued on next page*

*Figure 1 continued*

with the zebrafish and human *preKiss* genes. The exon-intron data were obtained from the respective genomic sequences from NCBI (MRZV01001091.1, NC_007122.7, NC_007115.7 and NC_000001.11). DNA structure is shown with exons numbered in green bands. ATG represents the start methionine codon and TGA represents the stop codon. (C) Alignment of the predicted echinoderm kisspeptin core sequences and functionally characterized chordate kisspeptins. Sequences of *Holothuria scabra, Holothuria glaberrima, Strongylocentrotus purpuratus,* and *Asterias rubens* kisspeptins were predicted by Elphick's lab (*Semmens and Elphick, 2017*; *Suwansa-Ard et al., 2018*). Vertebrate kisspeptin core sequences were obtained from GenBank with detailed sequences listed in *Figure 1—source data 1*. The color align property was generated using Sequence Manipulation Suite online. The percentage of sequences that must agree for identity or similarity coloring was set as 40%. (D) Phylogenetic tree of the kisspeptin precursor and four different neuropeptide outgroups (*Mirabeau and Joly, 2013*). The tree was constructed on the basis of approximately Maximum-Likelihood algorithms using FastTree two with pre-trimmed sequences. Local support values (%) were provided using the Shimodaira-Hasegawa (SH) test and are indicated by numbers at the nodes. The detailed complete sequences are listed in *Figure 1—source data 2* and trimmed sequences are listed in *Figure 1—source data 3*.

The online version of this article includes the following source data and figure supplement(s) for figure 1:

**Source data 1.** Core sequences of kisspeptin from multiple species for alignment.
**Source data 2.** Amino-acid sequences of the kisspeptin precursor and outgroups for phylogenetic analysis.
**Source data 3.** Trimmed sequence alignment for phylogenetic tree construction.
**Figure supplement 1.** Gene structure of the *Apostichopus japonicus* kisspeptin precursor.

ORF region (*Figure 1B*). Alignment of multiple sequences revealed a high similarity between AjKiss1a/b and predicted echinoderm kisspeptins, but low identity between AjKiss1a/b and vertebrate kisspeptin 1/2 (*Figure 1C*). A maximum likelihood tree of kisspeptin precursors, including PrRP, 26RFa/QRFP, GnIH, and NPFF from outgroups (*Ukena et al., 2014*), was constructed for phylogenetic analysis. It showed that *AjpreKiss*, together with kisspeptin-like precursor genes from the sea cucumbers *Holothuria scabra* and *H. glaberrima*, were grouped with the vertebrate kisspeptin 1 and kisspeptin 2 subfamilies into the 'Kisspeptin' group (*Figure 1D*).

Several predicted 'G-protein coupled receptor 54-like' or 'kisspeptin receptor-like' gene annotations in the hemichordate *Saccoglossus kowalevskii* (two genes), the echinoderm *Acanthaster planci* (two genes), and *S. purpuratus* (seven genes) have been reported (*Elphick, 2013*; *Simakov et al., 2015*; *Hall et al., 2017*). Using these predicted genes as reference sequences to search the *A. japonicus* genomic database, three *A. japonicus* kisspeptin receptor-like genes (*AjKissR1, AjKissR2*, and *AjKissRL3*; GenBank accession numbers, MH709114, MH709115, and MG199220, respectively) were identified and cloned from an *A. japonicus* ovary by RT-PCR. The ORFs of both *AjKissR1* and *AjKissR2* comprised three exons (*Figure 2A*), with deduced amino acid sequences of 378 and 327 residues, and contained seven transmembrane domains (*Figure 2B* and *Figure 2—figure supplement 1*). (Detailed data for another putative receptor *AjKissRL3* have not been presented because it exhibited no interaction with ligands in further experiments.) Sequence alignment of AjKissR1 and AjKissR2 with the well-characterized chordate GPR54 was performed (*Figure 2—figure supplement 2*) and a relatively high identity in seven-transmembrane region sequences, against 21 vertebrate GPR54 sequences, was observed (as shown in *Figure 2—figure supplement 3*). Maximum likelihood phylogenetic tree analysis, using 'Allatostatin-A receptor' and 'Galanin receptor' as outgroups, revealed that AjKissR1 and AjKissR2 both clustered in the 'Kisspeptin receptor' group. AjKissR1 grouped with *S. purpuratus* (sea urchin) kisspeptin receptors (XP_784787.2, XP_796690.2, XP_793873.2 and XP_796286.1) and hemichordate *S. kowalevskii* (acorn worm) kisspeptin receptor (NP001161574.1), whereas AjKissR2 clustered with the predicted *A. planci* (starfish) Kisspeptin receptors (Genbank ID: XP_022096858.1 and XP_022096775.1) and with the *S. purpuratus* kisspeptin receptor (XP_003727259.1) (*Figure 2C*).

## Functional expression of putative kisspeptin receptors

To verify the exact expression and localization of the putative *A. japonicus* kisspeptin receptors, AjKissR1 and AjKissR2 with an N-terminal FLAG-tag or with enhanced green fluorescent protein (EGFP) fused to the C-terminal end were constructed and stably or transiently expressed in human embryonic kidney 293 (HEK293) cells, respectively. As shown in *Figure 3A*, AjKissR1 and AjKissR2 were predominantly expressed and localized on the surface of HEK293 cells, with some intracellular accumulation, in the absence of the ligand. Next, to examine whether AjKissR1 and AjKissR2 are activated by synthetic kisspeptins, a $Ca^{2+}$ mobilization assay based on the calcium probe Fura 2 was

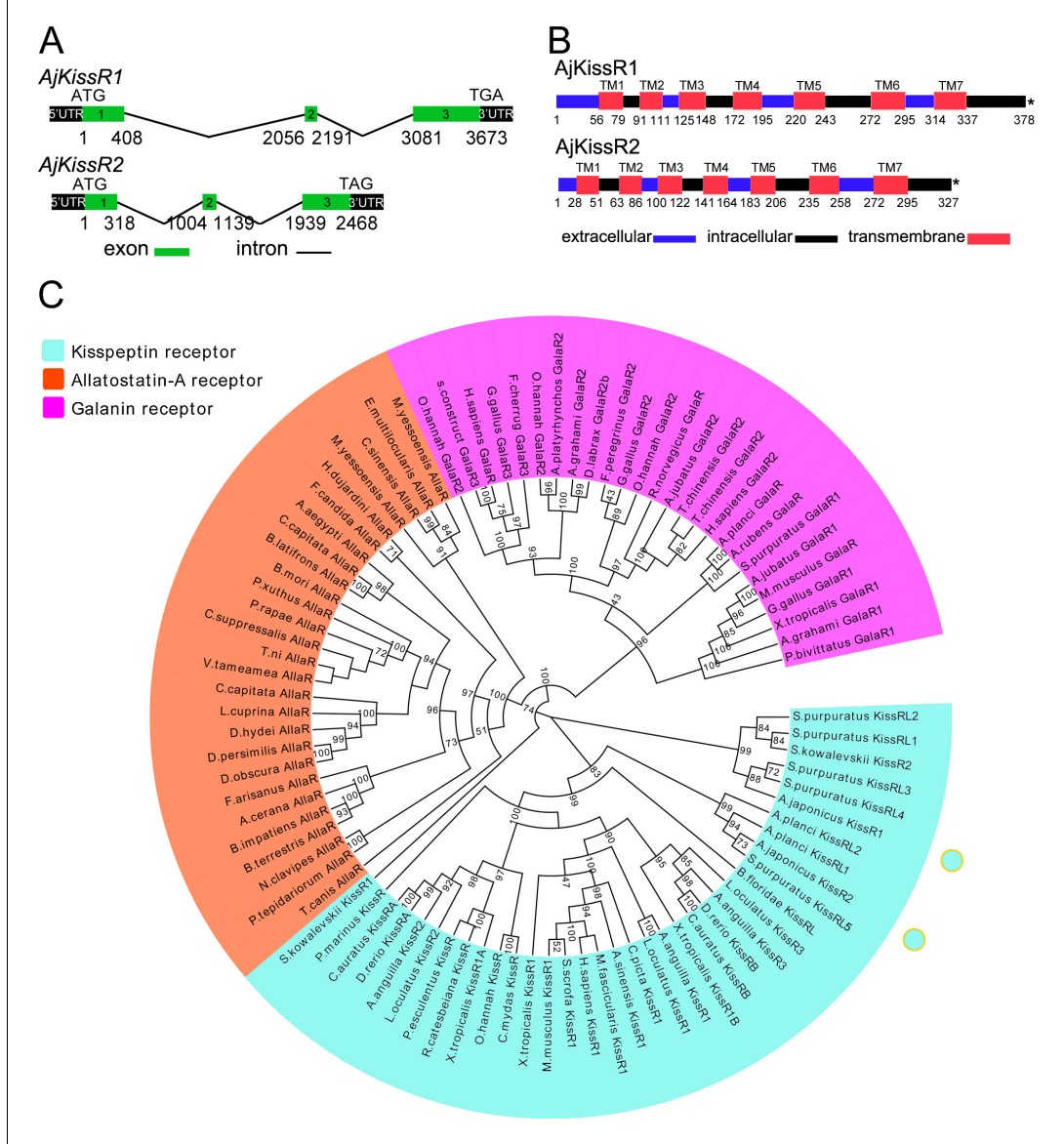

**Figure 2.** Gene structure and phylogenetic characterization of *Apostichopus japonicus* kisspeptin receptors (AjKissR1 and AjKissR2). (A) DNA structures of AjKissR1 and AjKissR2. *AjKissR1/2* DNA structure is shown with exons numbered in green bands. ATG represents the start methionine codon and TGA/TAG represents the stop codon. (B) Organization of the predicted protein structures. The seven transmembrane domains (TM1–TM7) are marked with red boxes. The N-terminal region and three extracellular (EC) rings are noted with blue boxes, and the C-terminal part and three intracellular (IC) rings are indicated with black boxes. Stop codons are represented by an asterisk. Arabic numbers under the bands indicate the nucleotide or amino acid sites. (C) Phylogenetic tree for kisspeptin, allatostatin-A and galanin receptors (KissR, AllaR and GalaR). The tree was constructed on the basis of approximately Maximum-Likelihood algorithms by FastTree two using AllaRs and GalaRs as outgroups (*Ukena et al., 2014*). Local support values were provided using the Shimodaira-Hasegawa (SH) test. The detailed sequences are listed in *Figure 2—source data 1*.

The online version of this article includes the following source data and figure supplement(s) for figure 2:

**Source data 1.** Amino-acid sequences of kisspeptin receptors and outgroups for phylogenetic analysis.

**Figure supplement 1.** Sequence, topology and annotations of *Apostichopus japonicus* kisspeptin receptors (A: AjKissR1, B: AjKissR2) visualized using the Protter webservice.

**Figure supplement 2.** Alignment of the deduced *Apostichopus japonicus* kisspeptin receptor amino-acid sequences with functionally characterized chordate GPR54 molecules from other species.

**Figure supplement 3.** Transmembrane region sequence similarity of *Apostichopus japonicus* kisspeptin receptors to vertebrate kisspeptin receptors.

**Figure supplement 3—source data 1.** Primary metadata of detailed identities for *Figure 2—figure supplement 3*.

performed. As shown in *Figure 3B*, both AjKiss1a and AjKiss1b elicited a rapid increase of intracellular $Ca^{2+}$, in a concentration-dependent manner, in HEK293 cells transfected with AjKissR1 and AjKissR2. However, AjKissR1 was preferentially activated by AjKiss1b, with an EC50 value of 8.06 nM (*Figure 3B 2*), whereas AjKissR2 was more specifically activated by AjKiss1a, with an EC50 value of 1.98 nM (*Figure 3B 1*).

Agonist-mediated receptor internalization from the cell surface into the cytoplasm has been recognized as a key mechanism in regulating the strength and duration of GPCR-mediated cell signaling and directly reflects the activation of the receptor (*Shenoy and Lefkowitz, 2003*; *Moore et al., 2007*). In this study, C-terminal fusion expression of AjKissR1 and AjKissR2 with EGFP was used to track the internalization and trafficking of receptors. As shown in *Figure 3C*, AjKissR1 and AjKissR2 were activated by AjKiss1b and AjKiss1a, respectively, to undergo significant internalization from the plasma membrane to the cytoplasm. These data provide clear evidence that AjKissR1 and AjKissR2 are functional receptors that are specific for neuropeptides AjKiss1b and AjKiss1a, respectively.

## Ligand selectivity of *A. japonicus* kisspeptin receptors

To examine the cross-activity of *A. japonicus* and vertebrate kisspeptin receptors, we detected the potential of synthetic *A. japonicus*, human, frog, and zebrafish kisspeptins (HsKiss1-10, XtKiss1b-10, DrKiss1-10, and DrKiss2-10) in triggering intracellular $Ca^{2+}$ mobilization. As indicated in *Figure 4*, HsKiss1-10 and XtKiss1b-10 exhibited higher potency for AjKissR1, whereas both DrKiss1-10 and

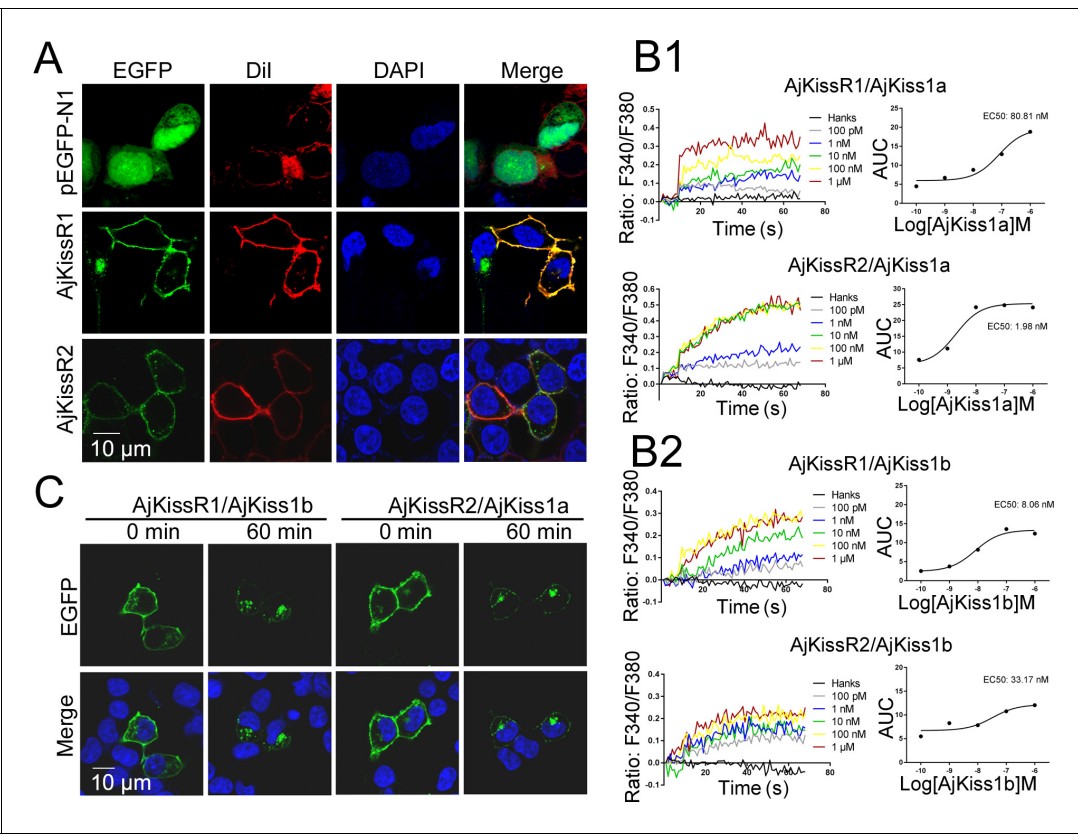

**Figure 3.** Functional characteristics of *Apostichopus japonicus* kisspeptins and receptors. (**A**) The cells transiently expressing AjKissR1-EGFP or AjKissR2-EGFP were stained with cell membrane probe (DiI) and cell nucleus probe (DAPI) and detected by confocal microscopy. (**B**) After loading with Fura-2/AM, HEK293 cells expressing either FLAG-AjKissR1 or FLAG-AjKissR2 were exposed to the indicated concentrations of AjKiss1a (B1) and AjKiss1b (B2), and continuous fluorescence was recorded. AUC, Area Under the Curve. *Figure 3B—source data 1* shows the primary metadata. (**C**) Internalization of AjKissR1-EGFP or AjKissR2-EGFP initiated by 1.0 μM of the indicated ligand was determined after a 60 min incubation by confocal microscopy. All pictures and data are representative of at least three independent experiments.

The online version of this article includes the following source data for figure 3:

**Source data 1.** Primary metadata of $Ca^{2+}$ mobilization assay for *Figure 3B*.

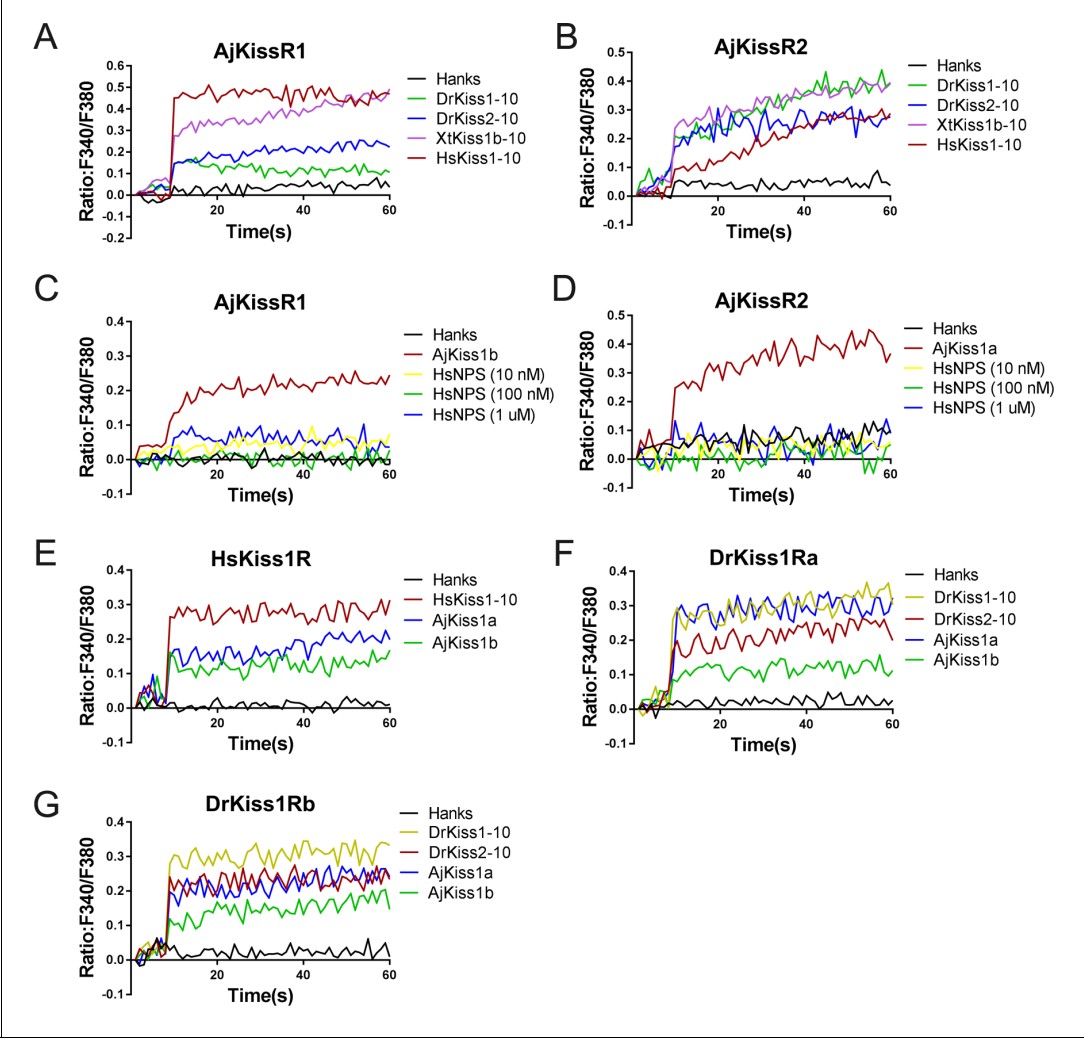

**Figure 4.** Functional cross-activity between the *A. japonicus* and vertebrate Kisspeptin/Kisspeptin receptor systems. Intracellular $Ca^{2+}$ mobilization in AjKissR1- (**A**) or AjKissR2-expressing (**B**) HEK293 cells was measured in response to 1.0 μM DrKiss1-10, DrKiss2-10, XtKiss1b-10, or HsKiss1-10 using Fura-2/AM. No $Ca^{2+}$-mobilization-mediated activity was detected in AjKissR1- (**C**) or AjKissR2-expressing (**D**) HEK293 cells upon administration of the indicated concentrations of human neuropeptide S (HsNPS). Intracellular $Ca^{2+}$ mobilization in human kisspeptin receptor (HsKiss1R)-expressing HEK293 cells was measured in response to 1.0 μM HsKiss1-10, AjKiss1a or AjKiss1b (**E**), as well as in zebrafish kisspeptin receptor (DrKiss1Ra or DrKiss1Rb)-expressing cells responding to 1.0 μM DrKiss1-10, AjKiss1a, or AjKiss1b (**F, G**). *Figure 4—source data 1* shows the primary metadata. All data shown are representative of at least three independent experiments.

The online version of this article includes the following source data for figure 4:

**Source data 1.** Primary metadata of $Ca^{2+}$ mobilization assay for *Figure 4A-G*.

DrKiss2-10 showed much lower potency in eliciting $Ca^{2+}$ mobilization (*Figure 4A*). For the activation of AjKissR2, however, XtKiss1b-10, DrKiss1-10, and DrKiss2-10 had a higher potency than that of HsKiss1-10 (*Figure 4B*). However, human neuropeptide S (HsNPS) showed no potency in activating either AjKissR1 or AjKissR2 (*Figure 4C and D*). Further analysis demonstrated that both AjKiss1a and AjKiss1b could activate HsKiss1R, DrKiss1Ra, and DrKiss1Rb with different potency (*Figure 4E, F and G*).

## *A. japonicus* kisspeptin receptors are directly activated by kisspeptins via a $G_{\alpha q}$-dependent pathway

Previous studies have demonstrated that in mammals, Kiss1R couples to $G_{\alpha q}$ protein, triggering phospholipase C (PLC), intracellular $Ca^{2+}$ mobilization, and the PKC signaling cascade in response to

agonists (*Castaño et al., 2009*). To elucidate G protein coupling in the activation of both AjKiss1a and AjKiss1b, a combination of functional assays, with different inhibitors, was performed. As shown in *Figure 5A*, AjKiss1a- and AjKiss1b-eliciting $Ca^{2+}$ mobilization through receptors AjKissR1 and AjKissR2, respectively, was completely blocked by pre-treatment with FR900359, a specific inhibitor of $G_{\alpha q}$ protein (*Lapadula et al., 2018*), and also significantly attenuated by PLC inhibitor U73122, the extracellular calcium chelator EGTA, and intracellular calcium chelator 1,2-bis(o-aminophenoxy) ethane N,N,N',N'-tetraacetic acid acetoxymethyl ester (BAPTA-AM) (*Shen et al., 2017*). By contrast, the $G_{\alpha i}$ protein inhibitor Pertussis Toxin (PTX) showed no effect on the $Ca^{2+}$ mobilization activated by *A. japonicus* kisspeptins in AjKissR1- or AjKissR2-expressing cells. These data indicate the involvement of $G_{\alpha q}$ protein in the AjKissR1- and AjKissR2-mediated intracellular signaling pathway.

Next, a competitive binding assay was established by using a synthesized AjKiss1a that had a fluorescein isothiocyanate (FITC) tag at the N-terminus (FITC–AjKiss1a, sequence listed in the 'Key Resources Table') to assess the direct interaction of AjKissR1 and AjKissR2 with AjKiss1a and AjKiss1b. Functional assays revealed that FITC–AjKiss1a exhibited the potential to induce $Ca^{2+}$ mobilization that was comparable to that induced by the wild-type neuropeptide (*Figure 5—figure supplement 1*). The competitive displacement of FITC–AjKiss1a with AjKiss1a and AjKiss1b in HEK293/AjKissR1 and HEK293/AjKissR2 cells was measured by FACS (fluorescence-activated cell sorting) analysis. As shown in *Figure 5B*, unlabeled AjKiss1a and AjKiss1b were found to compete with FITC-labeled AjKiss1a with $IC_{50}$ values of 95.16 and 353.30 nM in AjKissR2- and AjKissR1-transfected HEK293 cells, respectively.

## AjKissR1 and AjKissR2 are activated by AjKiss1b-10 and signal through the $G_{\alpha q}$-dependent MAPK pathway

As AjKiss1b-10 exhibited high potency in activating both AjKissR1 and AjKissR2 in HEK293 cells (*Figure 6—figure supplement 1*), it was used to conduct further in vitro and in vivo experiments. The previous results reveal that AjKissR1 and AjKissR2 can be activated by ligands and signals through $G_{\alpha q}$-dependent $Ca^{2+}$ mobilization; however, the detailed signaling pathway remained to be elucidated. To address this, different inhibitors were used to test intracellular ERK1/2 activation in HEK293 cells expressing AjKissR1 and AjKissR2, treated with Ajkiss1b-10. As shown in *Figure 6A*, stimulation with AjKiss1b-10 led to the activation of both AjKissR1 and AjKissR2, inducing significant ERK1/2 activation. Further assessment demonstrated that AjKissR1- or AjKissR2-mediated activation of ERK1/2 was significantly blocked by the PLC inhibitor U73122 (10 μM) and by the PKC inhibitor Gö6983 (1.0 μM) (*Figure 6B and C*). Moreover, using a PKC subtype translocation assay, we determined that PKCα, PKCβI, and PKCβII are involved in the activation of the MAPK pathway (*Figure 6D and E*). Overall, these results suggest that AjKissR1 and AjKissR2, once activated by ligand, can activate the $G_{\alpha q}$ family of heterotrimeric G proteins, leading to dissociation of the G protein subunits $G_{\beta \gamma}$ and the activation of PLC, leading to intracellular $Ca^{2+}$ mobilization. This, in turn, activates PKC (isoform α and β) and the MAPK cascade, particularly ERK1/2, via the $G_{\alpha q}$/PLC/PKC signaling pathway (*Figure 6F*).

## Physiological functions of the kisspeptin signaling system in *A. japonicus*

To further assess the physiological roles of the kisspeptin signaling system in *A. japonicus*, we examined the tissue distribution of *A. japonicus* kisspeptin and its receptor, using custom rabbit polyclonal antibodies (anti-AjKiss1b-10 and anti-AjKissR1 [details shown in the 'Key Resources Table']; antibody specificities were evaluated as shown in *Figure 7—figure supplement 1*). Tissue-specific western blot analysis revealed the expression of the kisspeptin precursor in the respiratory tree (RET), ovary (OVA), testis (TES), and anterior part (ANP, containing the nerve ring as shown in *Figure 7—figure supplement 2E,F*) of mature sea cucumbers (the maturity of the gonads was evaluated by H and E staining, as shown in *Figure 7—figure supplement 2B*). AjKissR1 was detected in the RET, OVA, ANP, and muscle (MUS) (*Figure 7A*). However, the failure to detect mature peptide fragment using anti-AjKiss1b-10 indicates that the further development of antibodies with high sensitivity and specificity might be required to clarify location of the mature kisspeptin in *A. japonicus* tissues.

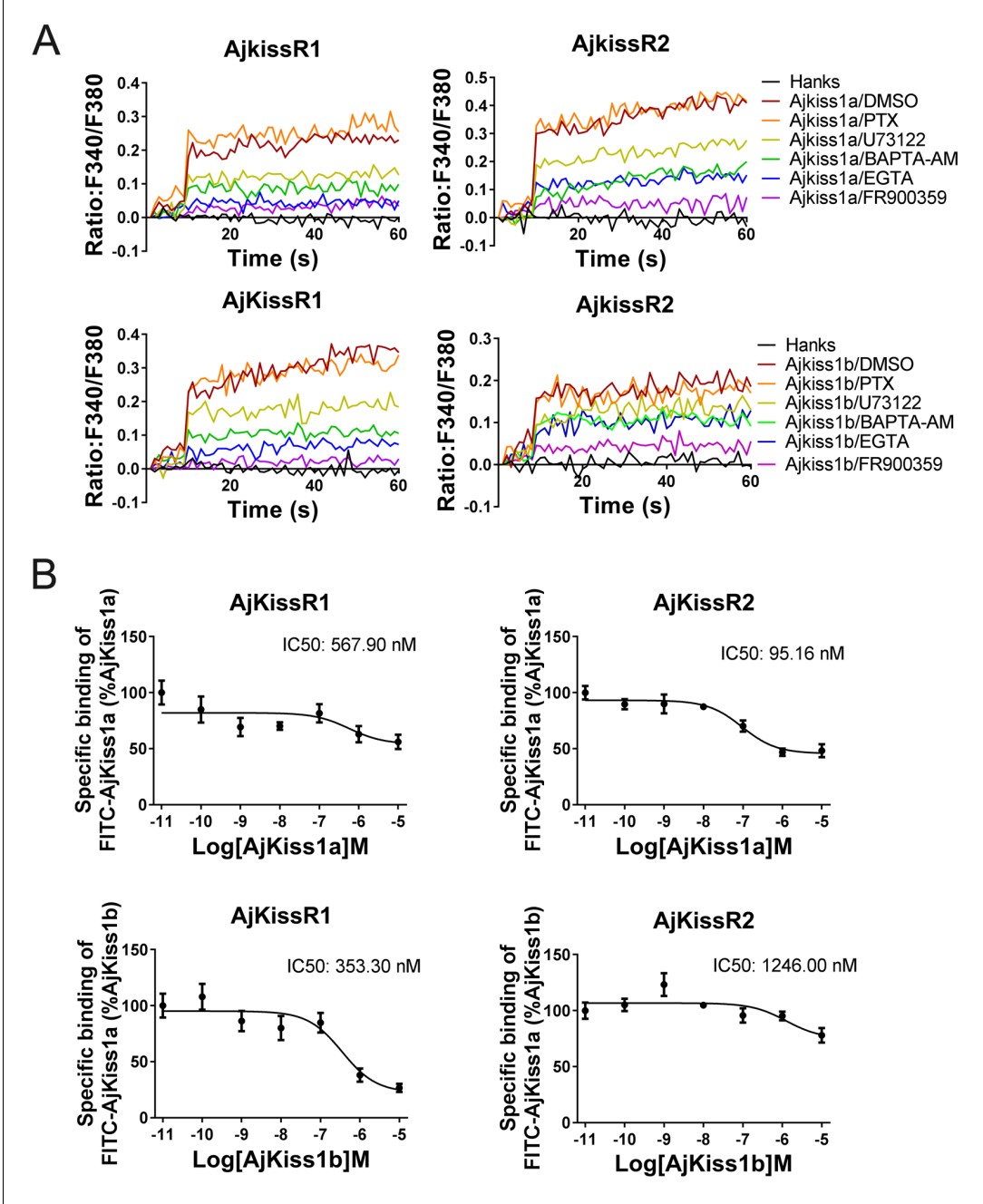

**Figure 5.** *Apostichopus japonicus* kisspeptin receptors are directly activated by kisspeptins via a $G_{\alpha q}$-dependent pathway. (**A**) Intracellular $Ca^{2+}$ mobilization in AjKissR1- and AjKissR2-expressing HEK293 cells was measured in response to 100 nM AjKiss1a or AjKiss1b, using cells that had been pre-treated for 12 hr with $G_{\alpha i}$ protein inhibitor (PTX, 100 ng/mL), or for 1 hr with DMSO, $G_{\alpha q}$ protein inhibitor (FR900359, 1.0 µM), PLC inhibitor (U73122, 10 µM), intracellular calcium chelator (BAPTA-AM, 100.0 µM), or extracellular calcium chelator (EGTA, 5.0 mM). *Figure 5— source data 1* presents the primary metadata. Pictures shown are representative of at least three independent experiments. (**B**) Competitive binding of 1.0 µM fluorescein isothiocyanate (FITC)-AjKiss1a to AjKissR1 or AjKissR2 in the presence of the indicated concentration of AjKiss1a or AjKiss1b. Error bars represent the SEM for at least three independent experiments.

The online version of this article includes the following source data and figure supplement(s) for figure 5:

**Source data 1.** Primary metadata of $Ca^{2+}$ mobilization assay and binding assay for *Figure 5A and B*.

**Figure supplement 1.** Functional activity of FITC-AjKiss1a evaluated by intracellular $Ca^{2+}$ mobilization detection.

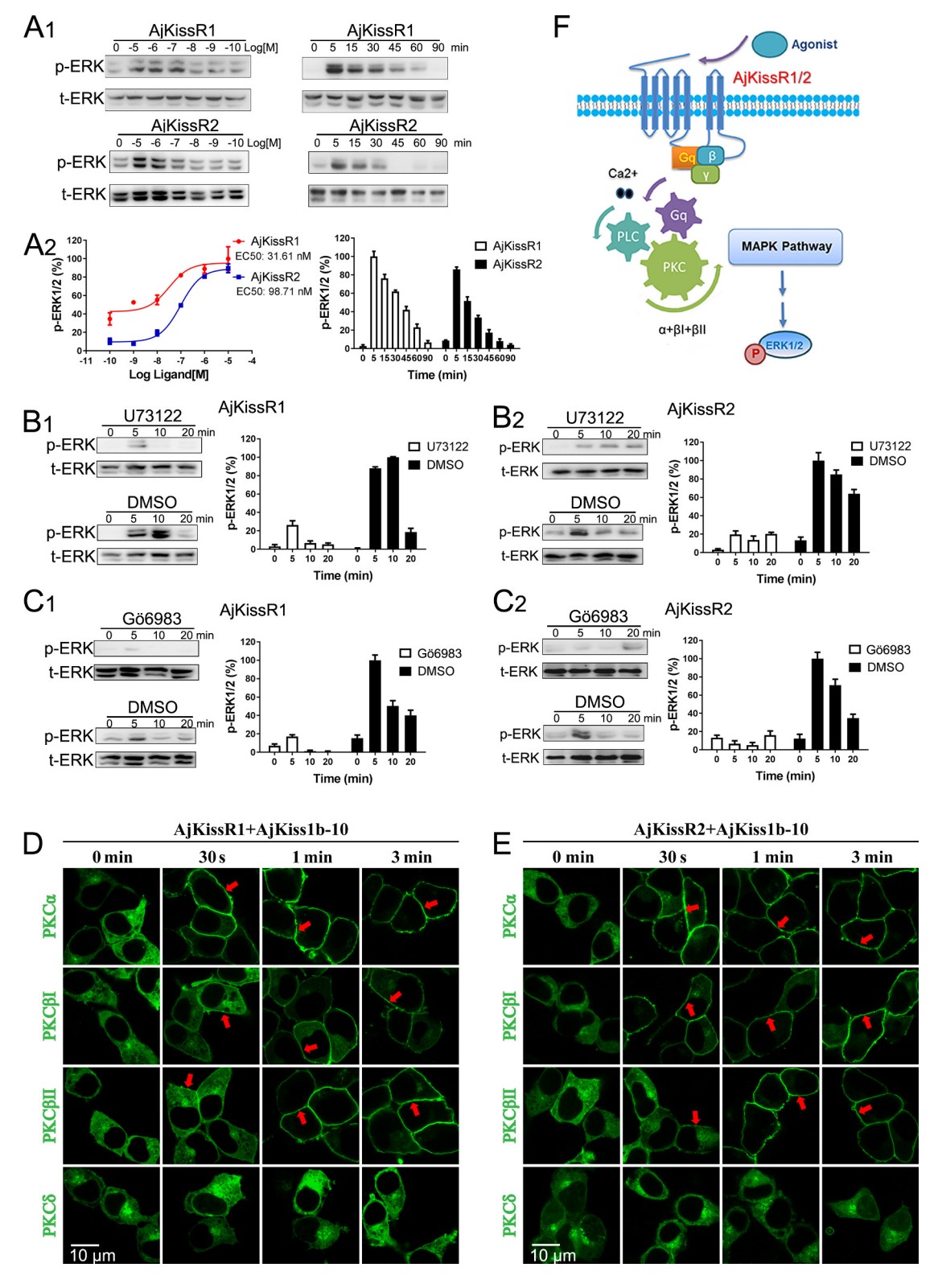

**Figure 6.** ERK1/2 activation mediated by AjKissR1 or AjKissR2. (**A**) Concentration- and time-dependent effects of AjKiss1b-10 on ERK1/2 phosphorylation in HEK293 cells that were stably expressing FLAG–AjKissR1 or FLAG–AjKissR2. Cells were challenged with different concentrations of AjKiss1b-10 for 5 min or incubated with 1.0 μM AjKiss1b-10 for the indicated times. Immunoblots were quantified using a Bio-Rad Quantity One Imaging system. (**B, C**) ERK1/2 phosphorylation, activated by AjKiss1b-10, was blocked by PLC or PKC inhibitors. Serum-starved HEK293 cells

*Figure 6 continued on next page*

*Figure 6 continued*

expressing FLAG–AjKissR1 or FLAG–AjKissR2 were pre-treated with DMSO, PLC inhibitor (U73122, 10 µM), or PKC inhibitor (Gö6983, 10 µM) for 1 hr prior to AjKiss1b-10 stimulation (1.0 µM). (**D, E**) Role of various PKC isoforms in the activated signaling pathways of the sea cucumber kisspeptin receptor. HEK293 cells, co-transfected with FLAG–AjKissR1 or FLAG–AjKissR2 and different PKC-EGFP isoforms, were stimulated by 1.0 µM AjKiss1b-10 for the indicated times and then examined by confocal microscopy. Red arrows denote the recruitment of PKC–EGFP isoforms on the cell membrane. (**F**) Schematic diagram of agonist-induced *A. japonicus* kisspeptin receptor activation. AjKiss1b-10 binding to AjKissR1 or AjKissR2 activates the $G_{\alpha q}$ family of heterotrimeric G proteins, which leads to dissociation of the G protein subunits $G_{\beta \gamma}$, and activates PLC, leading to intracellular $Ca^{2+}$ mobilization. This, in turn, activates PKC (isoform α and β) and stimulates the phosphorylation of ERK1/2. The ratio of p-ERK1/2 to total ERK1/2 was normalized to the peak value detected in the corresponding experiments (for example, the peak value of the ratio of AjKissR1/AjKiss1b-10 (10 µM) for the dose-dependent analysis, or of AjKissR1/AjKiss1b-10 (1.0 Aj) at 5 min for the time-course analysis). All pictures and data are representative of at least three independent experiments.

The online version of this article includes the following figure supplement(s) for figure 6:

**Figure supplement 1.** Functional activity of AjKiss1b-10.

To reveal the in situ distribution of the kisspeptin precursor and receptor, we performed immunofluorescence labeling on tissue sections. Consistent with results from the western blot assay, significant expression of the kisspeptin precursor was observed in the RET, TES, and nerve ring in the ANP sections, with no expression in the MUS and OVA (the inconsistency with results from the western blotting assay vs. immunofluorescence may be due to either differences in the specific structures and components of oocytes or differences in the characteristics of the antibody); AjKissR1 expression was observed in the RET, OVA, MUS and nerve ring in ANP sections, with rare expression in TES (*Figure 7B*). At the cellular level, the kisspeptin precursor was mainly detected in the coelomic epithelium of RET, whereas AjKissR1 was detected in the brown bodies, which can be found in the luminal spaces of RET and might be related to foreign material removal (*Smiley, 1994*). In particular, significant expression and cell membrane localization of AjKissR1 was detected in oocytes, indicating the consistent molecular property of AjKissR1 in vivo and in vitro. From the TES sections, significant fluorescence signal from the kisspeptin precursor, but only a weak signal from AjKissR1, can be detected in spermatogenic epithelium. A significant expression of AjKissR1 was detected in the epithelium of muscle from MUS sections. Moreover, in the ANP sections, the kisspeptin precursor was detectable in the outer surface part of the nerve ring (mainly containing the cell bodies of neurons, as shown in *Figure 7—figure supplement 2 F2*), whereas the AjKissR1 was detected in the internal region of the nerve ring (mainly containing the axons of neurons, as shown in *Figure 7—figure supplement 2 F2*). The distribution profile of *A. japonicus* kisspeptin and kisspeptin receptor indicates the potential roles of the kiss signaling system in physiological processes, such as reproduction, nervous system activity and metabolism.

To verify the physiological function of *A. japonicus* kisspeptins, cultured oocytes were stimulated by different kisspeptins. As shown in *Figure 7C*, significant ERK phosphorylation signal can be detected by western blot assay in different kisspeptin-treated oocytes. This signal can be blocked by the kisspeptin antagonist pep234 (1.0 µM) in DrKiss1-10 or AjKiss1b-10 administrated cells (the inhibitory effect of pep234 was validated in vitro as shown in *Figure 7—figure supplement 3*). Further, detection by confocal microscopy of the p-ERK signal in treated oocytes demonstrated the activation of this pathway by AjKiss1b-10 and its inhibition by pep234 in *A. japonicus* cells (*Figure 7D*).

Having confirmed their functional activity in cultured oocytes, AjKiss1b-10 and pep234 were used to conduct further in vivo experiments. Sea cucumbers treated with AjKiss1b-10 for 40 days exhibited weight loss (p=0.0583, Tukey's multiple comparisons test, as shown in *Figure 7E*) and extremely significant intestinal degeneration (p=0.0001, Tukey's multiple comparisons test, as shown in *Figure 7F,G*), which are characteristic phenotypes of aestivating *A. japonicus* (*Wang et al., 2015*). Moreover, extremely significant elevation of transcription of the gene encoding pyruvate kinase (PK) (p=0.0002, PBS vs. AjKiss1b-10, Tukey's multiple comparisons test, as shown in *Figure 7—figure supplement 4A*), which is the rate-limiting enzyme in the regulation of *A. japonicus* glycolysis (*Xiang et al., 2016*), was detected in the respiratory tree, whereas a significant decrease of PK transcription was found in the intestine (p=0.0188, PBS vs. AjKiss1b-10, Tukey's multiple comparisons test, as shown in *Figure 7—figure supplement 4A*). These data suggest that the *A. japonicus* kisspeptin system plays a role in the control of metabolic balance. To evaluate the potential role of

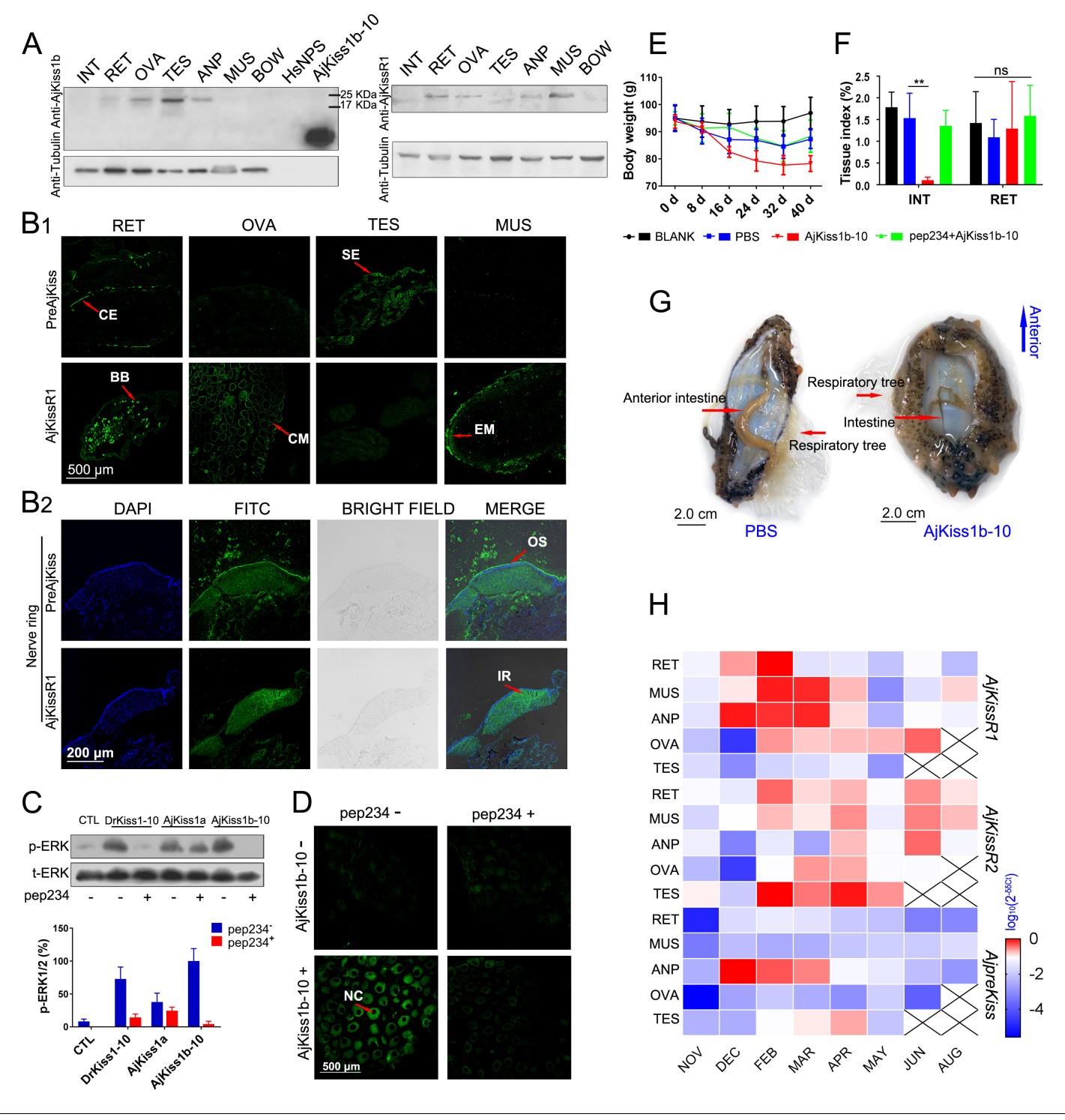

**Figure 7.** Physiological function analysis of kisspeptin signaling systems in *Apostichopus japonicus*. (**A**) Western Blot analysis of *A. japonicus* kisspeptin precursor and kisspeptin receptor (AjKissR1) in different tissues of sea cucumber. INT, intestine; RET, respiratory tree; ANP, anterior part; OVA, ovary; TES, testis; MUS, muscle; and BOW, body wall. (**B**) Immunofluorescence histochemical staining of *A. japonicus* kisspeptin precursor and AjKissR1 in RET, OVA, TES, MUS (B1) and nerve ring (B2) of the sea cucumber. CE, coelomic epithelium; BB, brown body; CM, cell membrane; SE, spermatogenic epithelium; EM, epithelium of muscle; OS, outer surface; and IR, internal region. (**C**) ERK1/2 phosphorylation activity of kisspeptins and the inhibitory effect of a vertebrate kisspeptin antagonist (pep234) on the cultured ovary of sea cucumber. Samples were collected and fixed after 2 hr of ligand administration with or without a 4-hr pre-treatment with pep234, in optimized L15 medium at 18°C. Error bars represent SEM for three independent experiments. Immunoblots were quantified using a Bio-Rad Quantity One Imaging system. (**D**) Immunofluorescence histochemical staining of p-ERK

*Figure 7 continued on next page*

*Figure 7 continued*

signal in cultured oocytes of sea cucumber. Samples were collected and fixed after 2 hr of ligand administration with or without a 4-hr pretreatment with pep234, in optimized L15 medium at 18°C. NC indicates the nucleus of oocytes. (**E, F**) Variation of body weight and tissue index (ratio of tissue weight/body weight) over 40 days of stimulus treatment. Each symbol and vertical bar represent mean ± SEM (n = 5 animals). ** indicates extremely significant differences (p=0.0001), as demonstrated by one-way ANOVA followed by Tukey's multiple comparisons test. (**G**) Degenerated intestine in AjKiss1b-10 treated sea cucumbers. (**H**) Heatmap showing the expression profile of *A. japonicus* kisspeptin and kisspeptin receptors (*AjKissR1/R2* and *AjKiss1*) in different tissues and developmental stages of sea cucumber. The variation in color represents the relative expression level of each gene in different samples (normalized against the peak values in all samples and logarithmized). The number of animals used for all samples is six, except for the number of ovary samples, with one in NOV (November) and JUN (June), three in DEC (December) and FEB (February), five in MAR (March), and six in APR (April) and MAY (May), and in testis, with two in NOV (November) and DEC (December), four in FEB (February) and MAR (March), and six in APR (April) and MAY (May). *Figure 7—source data 1* represents the primary metadata. All pictures and data are representative of at least three independent experiments.

The online version of this article includes the following source data and figure supplement(s) for figure 7:

**Source data 1.** Primary metadata of body weight, tissue index and qPCR assay for *Figure 7E, F and H*.
**Figure supplement 1.** Antibody specificities of anti-AjKiss1b-10 and anti-AjKissR1 IgG antibodies.
**Figure supplement 2.** General morphology and histology of *Apostichopus japonicus* tissues.
**Figure supplement 3.** Inhibitory effect of pep234 on AjKissR1 and AjKissR2 activation.
**Figure supplement 4.** Functional activity of AjKiss1b-10 in *Apostichopus japonicus*.
**Figure supplement 4—source data 1.** Primary metadata of qPCR assay and E2 concentration for *Figure 7—figure supplement 4A and B*.
**Figure supplement 5.** Mean body weight (**A**) and tissue index (**B**) change over annual investigation.
**Figure supplement 5—source data 1.** Primary metadata of body weight and tissue index in annual investigation for *Figure 7—figure supplement 5A and B*.

AjKiss1b-10 in regulating reproductive activity, we examined the estradiol (E2) levels in the coelomic fluid of sea cucumber, but no significant difference was observed in animals treated with AjKiss1b-10 (*Figure 7—figure supplement 4B*).

The transcriptional expression of the *A. japonicus* kisspeptin precursor (*AjpreKiss*) and of the kisspeptin receptors (*AjKissR1/2*) was investigated at different stages of reproductive development using the qPCR method. Two-year-old sea cucumbers, with 85.29 ± 9.47 g body weight (*Figure 7—figure supplement 5A*), were collected and various tissues were sampled for further analysis. As shown in *Figure 7—figure supplement 5B*, notable changes in the relative gut mass and the relative ovary weight of the sea cucumbers were detected in the developing reproductive stage from November to April, the mature reproductive stage in May, after spawning in June, and during aestivation in August. At all stages, *AjKissR1/2* expression was detectable in the majority of sea cucumber tissues, especially after February (*Figure 7H*), whereas significant expression of *AjpreKiss* was found in the ANP from December to April with a peak value detected in February. Taken together, the high expression levels of *AjpreKiss* during reproductive development suggests its role in the regulation of seasonal reproduction, whereas the wide distribution of *AjKissR1* and *AjKissR2* in the other tissues investigated indicates diverse functions for these two receptors.

## Discussion

The functional characterization of neuropeptides or secretory neurons of non-vertebrates contributes to our understanding of the evolutionary origin and conserved roles of the neurosecretory system in animals, especially in Ambulacrarians (deuterostomian invertebrates including hemichordates and echinoderms), which are closely related to chordates (*Tessmar-Raible et al., 2007*; *Odekunle et al., 2019*). The hypothalamic neuropeptide kisspeptin acts as a neurohormone and plays important roles in the regulation of diverse physiological processes in vertebrates, including reproductive development (*Popa et al., 2008*; *Franssen and Tena-Sempere, 2018*), metastasis suppression (*Ciaramella et al., 2018*), metabolism and development (*Song et al., 2014*; *Jiang et al., 2017*; *Katugampola et al., 2017*), behavioral and emotional control (*Comninos et al., 2017*), and the innate immune response (*Huang et al., 2018*).

A functional kisspeptin signaling system has been demonstrated in the chordate amphioxus (*Wang et al., 2017*), and a number of invertebrate kisspeptin genes have been predicted recently (*Mirabeau and Joly, 2013*; *Elphick and Mirabeau, 2014*; *Semmens et al., 2016*; *Semmens and Elphick, 2017*; *Suwansa-Ard et al., 2018*; *Chen et al., 2019*), however, missing experimental

identification of a kisspeptin-type system in non-chordates makes it difficult to determine whether this signaling system has an ancient evolutionary origin in invertebrates or whether it evolved de novo in the chordate/vertebrate lineages. In this study, two kisspeptin receptors from the sea cucumber *A. japonicus*, AjKissR1 and AjKissR2, have been identified as having a high affinity for synthetic kisspeptins from *A. japonicus* and vertebrates, and as sharing a similar $G_{\alpha q}$-dependent PLC/PKC signaling pathway with the mammalian kisspeptin signaling system. Results from the in vivo investigation indicate that the kisspeptin system in sea cucumber might be involved in the control of both metabolism and reproduction. Given the highly conserved intracellular signaling pathway and physiological functions revealed for the *A. japonicus* kisspeptin system, it is more likely that kisspeptin signaling might have originated from non-chordate invertebrates.

## Two putative kisspeptin receptors can be activated by multiple synthetic kisspeptin-type peptides in *A. japonicus*

Kisspeptins or kisspeptin receptors in Chordata have been functionally recognized in various species. Virtual screening of the transcriptome and genome sequence data for neuropeptide precursors has made a great contribution to the prediction of kisspeptin and its receptor paralogous genes in Ambulacrarians and has provided valuable information for further investigation (*Figure 8*). In 2013, kisspeptin-type receptors were first annotated in the genome of the acorn worm *S. kowalevskii* and the purple sea urchin *S. purpuratus* (*Jékely, 2013*; *Mirabeau and Joly, 2013*). Moreover, a kisspeptin-type neuropeptide precursor with 149 amino-acid residues was identified in the starfish *Asterias rubens*, comprising two putative kisspeptin-type peptides, ArKiss1 and ArKiss2 (*Semmens et al., 2016*). Subsequently, in silico analysis of neural and gonadal transcriptomes enabled the virtual discovery of kisspeptins in the sea cucumbers *H. scabra* and *H. glaberrima* (*Suwansa-Ard et al., 2018*). Moreover, the presence of kisspeptin-type peptides in extracts of radial nerve cords was confirmed by proteomic mass spectrometry in the crown-of-thorns starfish *A. planci* (*Smith et al., 2017*). Recently, a 180-residue protein comprising two putative kisspeptin-type peptides has been predicted and a C-terminally amidated peptide GRQPNRNAHYRTLPF-NH$_2$ was confirmed by mass spectrometric analysis of central nerve ring extracts of *A. japonicus* (*Chen et al., 2019*). These advances provide a basis for experimental studies on the kisspeptin signaling system in echinoderms.

In the present study, we cloned the full-length of *Kiss* cDNA sequence from the ANP tissue samples, encoding a putative kisspeptin precursor, which has been predicted from the proteomic analysis of *A. japonicus* (*Chen et al., 2019*) and synthesized the peptides AjKiss1a (32aa), AjKiss1a-15, AjKiss1a-13, AjKiss1a-10, AjKiss1b (18aa), and AjKiss1b-10, for further experimental characterization. Two candidate *A. japonicus* kisspeptin receptors were screened from genomic data and cloned from ovary tissue, on the basis of the sequence of the identified kisspeptin receptors (*Biran et al., 2008*; *Elphick, 2013*; *Jékely, 2013*; *Mirabeau and Joly, 2013*; *Simakov et al., 2015*; *Hall et al., 2017*; *Wang et al., 2017*) and functionally characterized. Our data show that despite a low percentage homology between AjKissR1 and AjKissR2, both the receptors were efficiently activated by synthetic *A. japonicus* kisspeptin peptides (AjKiss1a and AjKiss1b), thereby triggering extensive $Ca^{2+}$ mobilization and initiating significant receptor internalization, albeit with a different potency, when expressed in a mammalian cell line. This is consistent with previous studies demonstrating that in non-mammalian species, synthetic Kiss1 and Kiss2 activated kisspeptin receptors in vitro with differential ligand selectivity (*Ohga et al., 2013*; *Lee et al., 2009*). In particular, the truncated peptide AjKiss1b-10 exhibited a high activity in eliciting intracellular $Ca^{2+}$ mobilization in AjKissR1/2-expressing HEK293 cells, whereas the truncated peptides, AjKiss1a-15, AjKiss1a-13, and AjKiss1a-10, failed to activate the receptors. The functional activity of the truncated peptide AjKiss1b-10 is not unusual, considering that alternative cleavage occurs in the kisspeptin peptides of vertebrates (*Kotani et al., 2001*; *Lee et al., 2009*); however, the inactivity of AjKiss1a-15, which has been identified from mass spectrometric detection in *A. japonicus* (*Chen et al., 2019*), raises more questions about the functional and structural characteristics of this neuropeptide and requires further investigation.

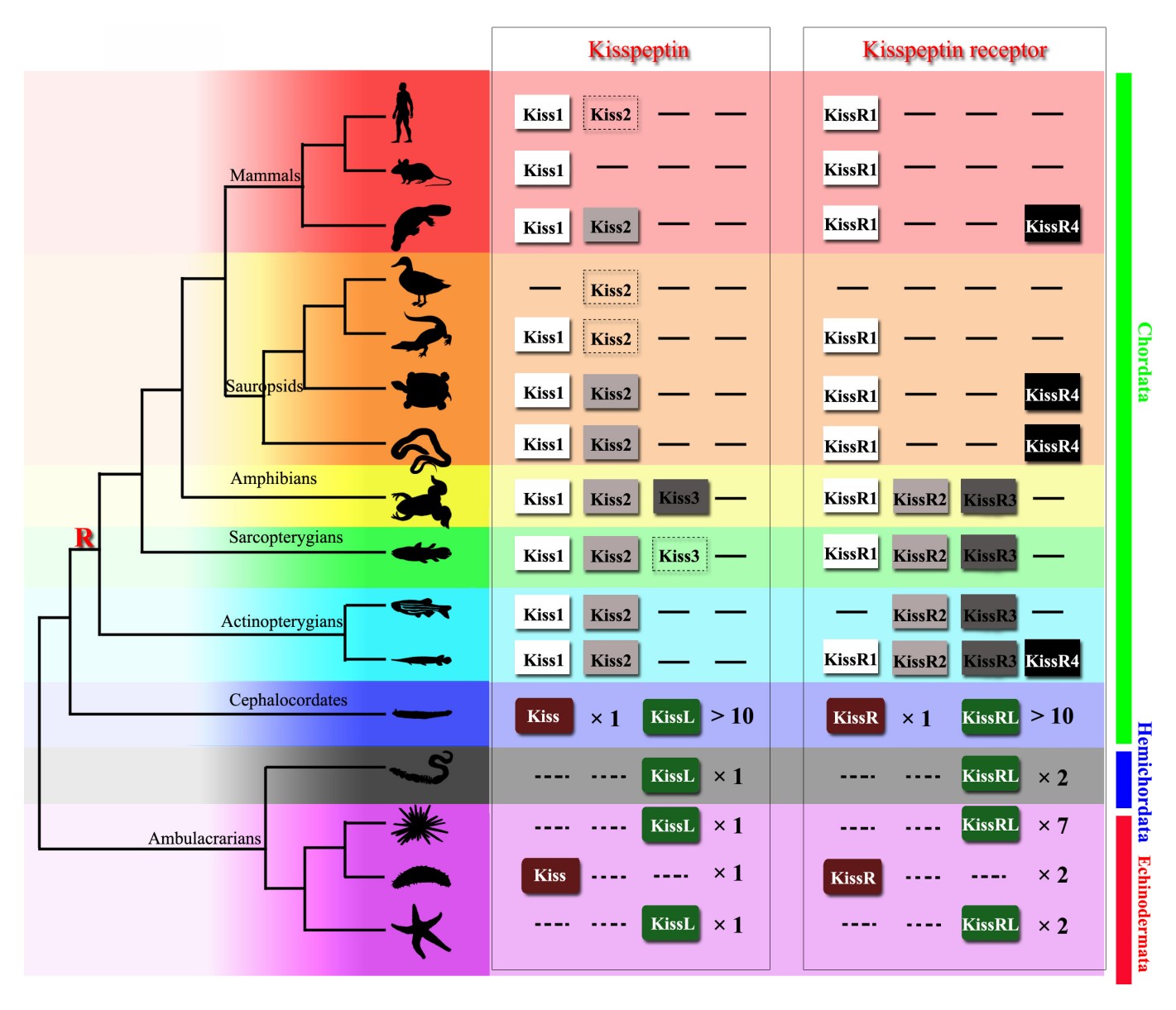

**Figure 8.** Recently identified Kisspeptin or Kisspeptin receptor genes among some deuterostomes. The species, indicated by silhouette images downloaded from the *PhyloPic* database, were clustered in a phylogenetic tree and classified by different colors. Red highlighted 'R' indicates a whole-genome duplication event. Kiss/KissR indicates the identified Kisspeptin/Kisspeptin receptor gene, and KissL/KissRL indicates a predicted Kisspeptin-like/Kisspeptin-like receptor gene. Dashed symbols indicate pseudogenes. Arabic numerals indicate the number of genes identified or predicted from public data. The evolutionary tree of the indicated species was modified from *Pasquier et al., 2014*. Image credits: all silhouettes from PhyloPic, human by T Michael Keeseyacorn; mouse by Anthony Caravaggi; platypus by Sarah Werning; duck by Sharon Wegner-Larsen; crocodile by B Kimmel; turtle by Roberto Díaz Sibaja; python by V Deepak; frog uncredited; coelacanth by Yan Wong; zebrafish by Jake Warner; spotted gar by Milton Tan; Branchiostoma by Mali'o Kodis, photograph by Hans Hillewaert; acorn worm by Mali'o Kodis, drawing by Manvir Singh; starfish by Hans Hillewaert and T Michael Keesey; sea cucumber by Lauren Sumner-Rooney; sea urchin by Jake Warner.

## Cross interaction of the kisspeptins and receptors between *A. japonicus* and vertebrates confirmed the existence of kisspeptin signaling systems in echinoderms

In the mammalian genome, a single *Kiss1* gene produces a mature 54-amino-acid peptide, Kiss-54, which is further proteolytically truncated to 14- and 13-amino-acid carboxyl-terminal peptides, Kiss-14 and Kiss-13, with a common C-terminal decapeptide (Kiss-10) core (*Kotani et al., 2001*;

*Ohtaki et al., 2001*). In non-mammalian vertebrates, two paralogous kisspeptin genes, *Kiss1* and *Kiss2*, are present in the genome of teleosts, producing two mature peptides that share the highly conserved Kiss-10 region with mammalian kisspeptin peptides (*Biran et al., 2008*; *Lee et al., 2009*; *Zmora et al., 2012*). Unlike mammalian and non-mammalian vertebrates, in the sea cucumber *A. japonicus*, only one kisspeptin gene was annotated and isolated. However, sequence analysis revealed that the kisspeptin gene encodes a 180-amino-acid peptide precursor, which is proteolytically cleaved to two mature peptides, consistent with other kisspeptins identified in the phylum Echinodermata (*Semmens et al., 2016*; *Semmens and Elphick, 2017*; *Smith et al., 2017*). Both putative mature peptides have a C-terminal Leu-Pro-Phe-amide motif, instead of the Arg-Phe-amide motif common in vertebrate kisspeptins, and exhibit a much lower identity with vertebrate kisspeptin sequences. Thus, the cross interaction of kisspeptin peptides and receptors between *A. japonicus* and vertebrates was further evaluated in this study.

Our specificity analysis showed that human, frog, and zebrafish kisspeptins, HsKiss1-10, XtKiss1b-10, and DrKiss1-10 and DrKiss2-10, were potent in activating both AjKissR1 and AjKissR2, whereas the human neuropeptide S (HsNPS, as a negative control) showed no potency to activate AjKissR1 or AjKissR2. Likewise, neuropeptides AjKiss1a and AjKiss1b could potentiate $Ca^{2+}$ signaling by binding the human kisspeptin receptor HsKiss1R and the zebrafish kisspeptin receptors DrKiss1Ra/b, similar to the corresponding active decapeptides. This, to our knowledge, is the first experimental data directly confirming the connection between the kisspeptin systems of vertebrates and *A. japonicus*, therefore proving that this peptide system is present and active in non-chordate deuterostome species. Considering the high conservation of the neuropeptides in different echinoderms (*Semmens and Elphick, 2017*; *Zandawala et al., 2017*), our findings strongly suggest that the kisspeptin signaling system that exists in *A. japonicus* may be extend to other taxa in this phylum.

## Conserved $G_{\alpha q}$/PLC/PKC/MAPK intracellular pathway mediated by the *A. japonicus* kisspeptin system provides insights into the evolution of kisspeptin signaling

It is well established that in mammals, Kiss1R is a typical $G_{\alpha q}$-coupled receptor, triggering intracellular $Ca^{2+}$ mobilization and the PLC/PKC signaling cascade in response to agonists (*Kirby et al., 2010*). However, accumulating evidence shows that in teleosts, although both kisspeptin receptors preferentially activate the $G_{\alpha q}$-dependent PKC pathway, one of them is also capable of triggering the $G_{\alpha s}$-dependent PKA cascade in response to kisspeptin challenge (*Ohga et al., 2013*; *Biran et al., 2008*). Using CRE-Luc and SRE-Luc reporting assays, which help to discriminate between the AC/PKA and PLC/PKC signaling pathways, an amphioxus kisspeptin receptor was shown to trigger significant PKC but not PKA signaling when stimulated by two kisspeptin-type peptides using heterologous expression in cultured HEK293 cells (*Wang et al., 2017*).

In this study, our data showed that, upon synthetic peptide stimulation, both AjKissR1 and AjKissR2 induced a rapid and transient rise in intracellular $Ca^{2+}$ in a $G_{\alpha q}$ inhibitor FR900359-sensitive manner. Further functional characterization demonstrated that both AjKissR1 and AjKissR2 induced ERK1/2 activation via a $G_{\alpha q}$/PLC/PKC cascade. Although the $G_{\alpha s}$ protein has been implicated in the teleost kisspeptin receptors-mediated signaling pathway, we failed to collect distinct evidence to prove the involvement of $G_{\alpha s}$-dependent signaling in the *A. japonicus* kisspeptin system. Taken together, it is more likely that $G_{\alpha q}$-coupled signaling is highly conserved in the kisspeptin signaling systems from *A. japonicus* to mammals.

## Reproductive and metabolic regulatory functions identified in *A. japonicus* revealed the ancient physiological roles of the kisspeptin system

Diverse physiological functions of the kisspeptin system have been reported in vertebrate species. In mammals, it is widely established that the kisspeptin signaling system is essential for HPG axis regulation, leading to reproductive control. Furthermore, the hypothalamic kisspeptin neurons have been found to stimulate pituitary gonadotropin-releasing hormone neurons, which express the kisspeptin receptor, providing a neural pathway for the mammalian kisspeptin neuronal system (*Oakley et al., 2009*). In non-mammalian species, especially in teleosts, the reproductive function of the kisspeptin system is still controversial in light of the normal reproductive phenotypes observed in fish in the

absence of kisspeptins. A new theory has been proposed in which the nonreproductive functions beyond HPG regulation are the conserved roles of kisspeptins in vertebrates (*Tang et al., 2015*; *Nakajo et al., 2018*). Here, we applied multiple approaches to analyze the potential functions of the recently identified kisspeptin in *A. japonicus*, aiming to provide some insights into the ancient physiological roles of the kisspeptin system.

In the present study, the expressional distribution of the *A. japonicus* kisspeptin and its receptor proteins in multiple tissues suggests the involvement of the kisspeptin signaling system in the regulation of both reproductive and non-reproductive functions. The seasonal fluctuation of kisspeptin and receptors transcripts, consistent with the reproductive development process, indicates the possible functional involvement of this system in the control of seasonal reproduction in *A. japonicus*. Interestingly, the unequally expressed kisspeptin and receptor protein levels in gonads, comparatively high kisspeptin precursor level in testis, and high kisspeptin receptor protein levels in ovary tissue demonstrated in our study, suggests differential functions of the kisspeptin system in different genders of sea cucumber. Further investigation using both in vivo and in vitro experiments indicated a role for the kisspeptin signaling system in regulating metabolic balance and gut function in sea cucumber. Combining the feeding regulatory function of VP/OT-type neuropeptides characterized in echinoderm (*Odekunle et al., 2019*) and the cross-talk between kisspeptin and VP/OT neural systems (*Higo et al., 2016*; *Seymour et al., 2017*; *Nakajo et al., 2018*), we suggest that a role of kisspeptin signaling in the regulation of the VP/OT system may exist in echinoderms, and the possible interaction between these two systems and an evolutionarily conserved function of the kisspeptin system is worthy of further exploration.

# Materials and methods

### Key resources table

| Reagent type (species) or resource | Designation | Source or reference | Identifiers | Additional information |
|---|---|---|---|---|
| Cell line (*Homo sapiens*) | HEK293 cell line | The National Institutes of Health (Bethesda, MD) | RRID:CVCL_0045 | Cell line maintained in this lab. |
| Antibody | Anti-phospho-ERK1/2(Thr$^{202}$/Tyr$^{204}$) (monoclonal, rabbit) | Cell Signaling Technology | CAT#9101 RRID:AB_331646 | IF (1:2000) |
| Antibody | Anti-ERK1/2 antibody (monoclonal, rabbit) | Cell Signaling Technology | CAT#9102 RRID:AB_330744 | IF (1:2000) |
| Antibody | Anti-beta Tubulin (monoclonal, rabbit) | Beyotime | CAT#AF1216 | IF (1:2000) |
| Antibody | FITC-conjugated goat anti-rabbit IgG (polyclonal, goat) | Beyotime | CAT#A0562 | IF (1:500) |
| Antibody | Cy3-conjugated goat anti-rabbit IgG (polyclonal, goat) | Beyotime | CAT#A0516 | IF (1:500) |
| Antibody | HRP-conjugated goat anti-rabbit IgG (polyclonal, goat) | Beyotime | CAT#A0208 | IF (1:500) |
| Antibody | anti-AjKiss1b-10 IgG (polyclonal, rabbit) | ChinaPeptides | CNE180821096 | Antigen sequence: CSRARPPLLPF-NH2 IF (1:1000) |
| Antibody | anti-AjKissR1 Ser$^{150}$~Trp$^{174}$ IgG (polyclonal, rabbit) | Wuhan Transduction Bio | PC059 | Antigen sequence: SYTRYQFIIHPLKA RAEWTSARVWW IF (1:1000) |
| Sequence-based reagent | Primers for plasmid construction AjKissR1–EGFP | This paper | PCR PRIMER | FORWARD CGAATTCATGTTTGA CGAAATGTTC EcoR I |

*Continued on next page*

*Continued*

| Reagent type (species) or resource | Designation | Source or reference | Identifiers | Additional information |
|---|---|---|---|---|
| Sequence-based reagent | Primers for plasmid construction AjKissR1–EGFP | This paper | PCR PRIMER | REVERSE GTGGATCCCGAACGATA CGATTCTGTTC BamH I |
| Sequence-based reagent | Primers for plasmid construction FLAG–AjKissR1 | This paper | PCR PRIMER | FORWARD GGAATTCATGTTTGAC GAAATGTTC EcoR I |
| Sequence-based reagent | Primers for plasmid construction FLAG–AjKissR1 | This paper | PCR PRIMER | REVERSE CGGGATCCTCAAA CGATACGATTCTGTTC BamH I |
| Sequence-based reagent | Primers for plasmid construction AjKissR2–EGFP | This paper | PCR PRIMER | FORWARD CGAATTCATGGAC AGCCTCTCAGC EcoR I |
| Sequence-based reagent | Primers for plasmid construction AjKissR2–EGFP | This paper | PCR PRIMER | REVERSE CCGTCGACTGAGT TACAGTATTTGCTG SalI |
| Sequence-based reagent | Primers for plasmid construction FLAG–AjKissR2 | This paper | PCR PRIMER | FORWARD CCAAGCTTGGATGGA CAGCCTCTCAGCGTT Hind III |
| Sequence-based reagent | Primers for plasmid construction FLAG–AjKissR2 | This paper | PCR PRIMER | REVERSE CGGGATCCCGTGAG TTACAGTATTTGCTGCAT Bam HI |
| Sequence-based reagent | Primers for plasmid construction FLAG–HsKiss1R | This paper | PCR PRIMER | FORWARD CCAAGCTTGGATGC ACACCGTGGCTAC Hind III |
| Sequence-based reagent | Primers for plasmid construction FLAG–HsKiss1R | This paper | PCR PRIMER | REVERSE CGGGATCCTCAGAG AGGGGCGTTGTCCT Bam HI |
| Sequence-based reagent | Primers for qPCR assays *AjKissR1* | This paper | PCR PRIMER | FORWARD AGTGGACATCT GCAAGAGTATGG |
| Sequence-based reagent | Primers for qPCR assays *AjKissR1* | This paper | PCR PRIMER | REVERSE CTTCCTGCGTAAT GGTATCGGTA |
| Sequence-based reagent | Primers for qPCR assays *AjKissR2* | This paper | PCR PRIMER | FORWARD TCTCGTTGTTGT CTTGACGTTTG |
| Sequence-based reagent | Primers for qPCR assays *AjKissR2* | This paper | PCR PRIMER | REVERSE TCGTCTGAAGT TTTCTCCCATGA |
| Sequence-based reagent | Primers for qPCR assays *AjpreKiss* | This paper | PCR PRIMER | FORWARD CCTACTGTCAT TGCTCTGTGGAAC |
| Sequence-based reagent | Primers for qPCR assays *AjpreKiss* | This paper | PCR PRIMER | REVERSE CAAGGTCATCT TCGTCTTGTTCTC |
| Sequence-based reagent | Primers for qPCR assays *β-tubulin* | This paper | PCR PRIMER | FORWARD CACCACGTGG ACTCAAAATG |
| Sequence-based reagent | Primers for qPCR assays *β-tubulin* | This paper | PCR PRIMER | REVERSE GAAAGCCTTACG ACGGAACA |

*Continued on next page*

*Continued*

| Reagent type (species) or resource | Designation | Source or reference | Identifiers | Additional information |
|---|---|---|---|---|
| Sequence-based reagent | Primers for qPCR assays *β-actin* | This paper | PCR PRIMER | FORWARD AAGGTTATGCT CTTCCTCACGC |
| Sequence-based reagent | Primers for qPCR assays *β-actin* | This paper | PCR PRIMER | REVERSE GATGTCACGGA CGATTTCACG |
| Recombinant DNA reagent | AjKissR1–EGFP | This paper | Plasmid | C-terminal EGFP-tag, backbone pEGFP-N1 |
| Recombinant DNA reagent | FLAG-–AjKissR1 | This paper | Plasmid | N-terminal FLAG-tag, backbone pCMV–FLAG |
| Recombinant DNA reagent | AjKissR2–EGFP | This paper | Plasmid | C-terminal EGFP-tag, backbone pEGFP-N1 |
| Recombinant DNA reagent | FLAG–AjKissR2 | This paper | Plasmid | N-terminal FLAG-tag, backbone pCMV–FLAG |
| Recombinant DNA reagent | FLAG–HsKiss1R | This paper | Plasmid | N-terminal FLAG-tag, backbone pCMV–FLAG |
| Recombinant DNA reagent | FLAG–DrKiss1Ra | This paper | Plasmid | N-terminal FLAG-tag, backbone pCMV–FLAG |
| Recombinant DNA reagent | FLAG–DrKiss1Rb | This paper | Plasmid | N-terminal FLAG-tag, backbone pCMV–FLAG |
| Recombinant DNA reagent | EGFP-tagged rat PKC isoforms (α, βI, βII and δ) | Kindly provided by Dr Jin O-Uchi, University of Rochester and Dr Naoaki Sato, Kobe University | Plasmid | C-terminal EGFP-tag, backbone pTB701 |
| Peptide, recombinant protein | AjKiss1a (C-terminal amidated) | This paper | | AGSLDc < CLEASC > EDVERRGRQPNRNA HYRTLPF-NH2 |
| Peptide, recombinant protein | FITC–AjKiss1a (C-terminal amidated) | This paper | | FITC–AGSLDc < CLEASC > EDVERRGRQPNRN AHYRTLPF-NH$_2$ |
| Peptide, recombinant protein | AjKiss1a-15 (C-terminal amidated) | This paper | | GRQPNRNAHYRT LPF-NH$_2$ |
| Peptide, recombinant protein | AjKiss1a-10 (C-terminal amidated) | This paper | | AHYRTLPF-NH$_2$ |
| Peptide, recombinant protein | AjKiss1b (C-terminal amidated) | This paper | | SAVKNKNKSRARP PLLPF-NH$_2$ |
| Peptide, recombinant protein | AjKiss1b-10 (C-terminal amidated) | This paper | | SRARPPLLPF-NH$_2$ |
| Peptide, recombinant protein | HsKISS1-10 (C-terminal amidated) | This paper | | YNWNSFGLRF-NH$_2$ |
| Peptide, recombinant protein | XtKISS3/KISS1b-10 (C-terminal amidated) | This paper | | YNVNSFGLRF-NH$_2$ |
| Peptide, recombinant protein | DrKISS1-10 (C-terminal amidated) | This paper | | YNLNSFGLRY-NH$_2$ |
| Peptide, recombinant protein | DrKISS2-10 (C-terminal amidated) | This paper | | FNYNPFGLRF-NH$_2$ |
| Peptide, recombinant protein | pep234 (C-terminal amidated) | This paper | | ac-(D-A)NWNGFG (D-W)RF-NH$_2$ |
| Commercial assay or kit | Rapid DNA Ligation kit | Beyotime | CAT#D7003 | |
| Commercial assay or kit | SYBR PrimeScript RT reagent Kit | TaKaRa | CAT #RR037A | |
| Commercial assay or kit | Iodine (125I) radioimmunoassay kit | Beijing North Institute of Biotechnology | S10940094 | |

*Continued on next page*

Continued

| Reagent type (species) or resource | Designation | Source or reference | Identifiers | Additional information |
|---|---|---|---|---|
| Chemical compound, drug | Pertussis toxin (PTX) | Tocris Bioscience | Cat #3097/50U | Specific inhibitor of $G_{\alpha i}$ |
| Chemical compound, drug | U73122 | Tocris Bioscience | Cat #1268/10 | PLC inhibitor |
| Chemical compound, drug | BAPTA-AM | Tocris Bioscience | Cat #2787/25 | Intracellular calcium chelator |
| Chemical compound, drug | EGTA | Tocris Bioscience | Cat #2807/1G | Extracellular calcium chelator |
| Chemical compound, drug | Gö6983 | Tocris Bioscience | Cat #2285/1 | Broad spectrum PKC inhibitor |
| Chemical compound, drug | FR900359 | Kindly provided by Dr Shihua Wu, Zhejiang University | | Specific inhibitor of $G_{\alpha q}$ |

## Animal collection and treatment

For cDNA cloning and gene expression analysis in various tissues, individuals of the sea cucumber *A. japonicus* were collected from separate culture ponds in Qingdao (Shandong, China, in 2016–2017). Each batch was acclimated in seawater aquaria (salinity range: 32.21–34.13) for 7 days and further dissected, sampled, and stored in liquid nitrogen for future use or directly used for tissue culture. Individuals for in vivo experiments (94 ± 4.3 g) were collected from the same culture pond in November 2017, kept in a 500 L tank, and fed with a formulated diet (45% marine mud, 50% Sargasso, and 5% shrimp shell powder) before chemicals were administered. After 15 days, sea cucumbers were randomly assigned to different groups (10 individuals per group). AjKiss1b-10 was dissolved in PBS and intraperitoneal injection of 100 µL AjKiss1b-10 (concentration of 0.5 mg/mL diluted in PBS) or of PBS alone was conducted once every 2 days, at noon. After 40 days (10 December 2017 to 18 January 2018) of chemical administration, animals were dissected and the respiratory tree, intestine, muscle, and anterior part tissues were taken as samples from five individuals for each group, and stored in liquid nitrogen for future use. Coelom fluid was collected and stored at −20℃ for E2 detection. This experiment was carried out on Xixuan Fishery Technology Island without temperature or light control (sea water temperature 11.5–7.0℃). Individuals used in the in vitro experiments (89 ± 2.4 g) were collected from the same culture pond in May 2017 and the respiratory tree, muscle, body wall, intestine, anterior part (containing the nerve ring), and ovary were dissected and further restored in −20℃ for western blotting or washed with PBS three times, in aseptic conditions, for tissue culture and in vitro experiments.

## Cell lines

HEK293 cells were provided from The National Institutes of Health (Bethesda, MD). Cells were authenticated by STR profiling and routinely tested for mycoplasma using commercial kits.

## Cell culture and transfection

HEK293 cells were cultured in Dulbecco's Modified Eagle's Medium (DMEM, HyClone) supplemented with 10% FBS (fetal bovine serum), 100 units/mL penicillin, 100 µg/mL streptomycin and 4.0 mM L-glutamine (Thermo Fisher Scientific) at 37℃ in a humidified incubator containing 5% $CO_2$. The plasmid constructs were transfected into HEK293 cells using X-tremeGENE HP (Roche), according to the manufacturer's instructions. Two days after transfection, stably expressing cells can be selected by the addition of 800 mg/L G418 to avoid impacts from inefficient transfection in experiments.

## Bioinformatic searches and tools

The cDNA sequences were used to query known sequences in GenBank using the blastx utility, BLASTX 2.8.0+ (http://blast.ncbi.nlm.nih.gov/). The cDNA sequence of *A. japonicus* kisspeptin precursor or kisspeptin receptors was translated into the predicted amino-acid sequence with DNA-MAN 8.0. Analysis of the physicochemical properties of proteins was based on Protparam (http://

[www.expasy.org/tools/protparam.html](www.expasy.org/tools/protparam.html)). Analysis of transmembrane regions in the protein was achieved by TMHMM ([http://topcons.cbr.su.se/](http://topcons.cbr.su.se/)). The deduced amino-acid sequences were aligned using ClustalW. The color-align property was generated by the Sequence Manipulation Suite ([http://www.bioinformatics.org/sms2/color_align_prop.html](http://www.bioinformatics.org/sms2/color_align_prop.html)). Signal peptide was predicted by SignalP-5.0 Server ([http://www.cbs.dtu.dk/services/SignalP/](http://www.cbs.dtu.dk/services/SignalP/)).

To construct the Maximum Likelihood (ML) phylogenetic tree, the Constraint-based Multiple Alignment Tool ([https://www.ncbi.nlm.nih.gov/tools/cobalt/cobalt.cgi?CMD=Web](https://www.ncbi.nlm.nih.gov/tools/cobalt/cobalt.cgi?CMD=Web)) was used to compute a multiple protein sequence alignment. The FastTree program (version 2.1) was then used to construct a tree based on the WAG and CAT model, which provided local support values that were based on the Shimodaira-Hasegawa (SH) test (*Price et al., 2010*). For the phylogenetic analysis of Kisses and outgroups, poorly aligned regions from the multiple alignment were automatically eliminated by trimAl (version 1.3, [http://trimal.cgenomics.org/](http://trimal.cgenomics.org/)) using the automated1 option (Heuristic Method) (*Capella-Gutiérrez et al., 2009*). According to the test of benchmark, the trimmed alignment always resulted in ML trees, which were of equal (36%) or significantly better (64%) quality than the tree produced by the complete alignment. The phylogenetic trees were visualized through the Evolview web server (*Subramanian et al., 2019*).

## Molecular cloning and plasmid construction

To construct the AjKissR1/2 fusion expression plasmids, RT-PCR was performed using total RNA extracted from *A. japonicus* ovaries to synthesize template cDNA. PCR amplification of the coding sequences of *AjKissR1/2* was performed using specific primers, with restriction sites (see 'Key Resources Table'). The corresponding PCR products were then cloned into pCMV–FLAG and pEGFP–N1 vectors using restriction enzymes and a Rapid DNA Ligation Kit (Beyotime, China). The FLAG–HsKiss1R plasmid was constructed using total synthesized DNA (Wuhan Transduction Bio) with specific primers containing restriction sites (see 'Key Resources Table'). All constructs were sequenced to verify the correct sequences, orientations, and reading frames.

## Intracellular calcium measurement

The fluorescent $Ca^{2+}$ indicator Fura-2/AM was used to detect intracellular calcium flux (*Li et al., 2010*). Briefly, the HEK293 cells expressing the AjKissR1, AjKissR2 or vertebrate kisspeptin receptor were washed twice with PBS and suspended at $5 \times 10^6$ cells/mL in Hanks' balanced salt solution. The cells were then loaded with 3.0 μM Fura-2/AM for 30 min and washed twice in Hanks' solution. They were then stimulated with the indicated concentrations of different predicted *A. japonicus* kisspeptins or vertebrate kisspeptins. Finally, intracellular calcium flux was measured for 60 s, by determining the ratio of excitation wavelengths at 340 and 380 nm using a fluorescence spectrometer (Infinite 200 PRO, Tecan, Männedorf, Switzerland). All of the experiments for measuring $Ca^{2+}$ mobilization were repeated independently at least three times.

## Receptor localization and translocation assay by confocal microscopy

For receptor-surface expression analysis, HEK293 cells expressing AjKissR1/2-EGFP were seeded onto glass coverslips in 12-well plates, coated with 0.1 mg/mL poly-L-lysine, and allowed to attach overnight under normal growth conditions (*Li et al., 2010*). The cells were washed three times with PBS and further stained with the membrane probe DiI (Beyotime) at 37°C for 5–10 min, fixed with 4% paraformaldehyde for 10 min, and then incubated with DAPI (Beyotime) for 5–10 min. For the translocation assays, the receptor-expressing cells were treated with 1.0 μM of various stimuli for 60 min, washed three times with PBS and then fixed with 4% paraformaldehyde in PBS for 10 min at room temperature. Finally, the cells were mounted in mounting reagent (DTT/PBS/glycerol, 1:8:2) and visualized by fluorescence microscopy on a Zeiss laser scanning confocal microscope, which was attached to a Zeiss Axiovert 200 microscope and linked to an LSM5 computer system.

## PKC translocation assay by confocal microscopy

For the translocation analysis of various PKC subtypes, HEK293 cells co-transfected with FLAG–AjKissR1 or FLAG–AjKissR2 and various GFP-tagged rat PKC isoforms (α, βI, βII and δ, see 'Key Resources Table') were seeded onto glass coverslips or six-well plates. After treatment with AjKiss1b-10 (1.0 μM) at 37°C for the indicated times, the cells were washed with PBS and fixed with

4% paraformaldehyde in PBS for 10 min at room temperature. Then, the cells were mounted in 50% glycerol and visualized by fluorescence microscopy using a Zeiss Axiovert 200 microscope linked to an LSM5 computer system. Excitation was performed at 488 nm, and the fluorescence detection used a 505–530 nm bandpass filter.

## Antibodies

The primary antibodies used for p-ERK1/2, ERK1/2, or β-tubulin detection were: rabbit anti-phospho-ERK1/2(Thr$^{202}$/Tyr$^{204}$) antibody (1:2000; Cell Signaling Technology), rabbit anti-ERK1/2 antibody (1:2000; Cell Signaling Technology), and beta-tubulin rabbit monoclonal antibody (1:2000; Beyotime). To examine the *A. japonicus* kisspeptin precursor or AjKissR1 in various tissues of sea cucumber, AjKiss1b-10 or a peptide corresponding to amino acids Ser$^{150}$~Trp$^{174}$ of AjKissR1, the second intracellular loop, was synthesized and injected into two rabbits, respectively. The polyclonal antibodies, rabbit anti-AjKiss1b-10 (1:1000) was prepared by ChinaPeptides, and anti-AjKissR1 (1:1000) was prepared by Wuhan Transduction Bio. The secondary antibodies used were HRP-conjugated goat anti-rabbit IgG, FITC-conjugated goat anti-rabbit IgG, and Cy3-conjugated goat anti-rabbit IgG (Beyotime).

## Protein extraction and western blotting

To examine the phosphorylation of ERK, cells that expressed AjKissr1/2 or other GPR54s were incubated for the indicated times with different concentrations of kisspeptins. Subsequently, cells were lysed with lysis buffer (Beyotime) that contained protease inhibitor (Roche) at 4°C for 30 min on a rocker and then scraped. Proteins were then electrophoresed on a 10% SDS polyacrylamide gel and transferred to polyvinylidene difluoride (PVDF) membranes. Membranes were blocked with 5% skimmed milk, then probed with rabbit anti-phospho-ERK1/2(Thr$^{202}$/Tyr$^{204}$) antibody (1:2000; Cell Signaling Technology), followed by detection using HRP-conjugated goat anti-rabbit IgG (Beyotime). Blots were stripped and reprobed using anti-ERK1/2 antibody (1:2000; Cell Signaling Technology) as a control for protein loading.

To detect AjKissR1 in different tissues of sea cucumber, the respiratory tree, intestine, muscle, nerve ring, and ovary were sampled and homogenized with lysis buffer (Beyotime) that contained protease inhibitor (Roche) at 4°C. Comparable concentrations of proteins were then electrophoresed on a 10% SDS polyacrylamide gel and transferred to PVDF membranes. Membranes were blocked with 5% skimmed milk, then probed with rabbit anti-AjKissR1 serum (1:1000), followed by detection using HRP-conjugated goat anti-rabbit IgG (Beyotime). Samples were probed in parallel with anti-tubulin antibody (Beyotime) as control for protein loading.

To detect AjpreKiss or its mature peptide in different tissues of sea cucumber, tissues were sampled and treated following the same protocol for receptor detection. Comparable concentrations of proteins were then electrophoresed on a 15% SDS polyacrylamide gel (preparation of 20 mL gel solution: 4.6 mL dH$_2$O, 10.0 mL 30% polyacrylamide solution, 5.0 mL 5 mol/L Tris [pH8.8], 0.2 mL 10% SDS solution, 0.2 mL 10% ammonium persulphate solution, 8.0 μL TEMED) and transferred to PVDF membranes (180 mA, 40 min for Kiss detection and 180 mA, 70 min for tubulin detection). Membranes were blocked with 5% skimmed milk, then probed with rabbit anti-AjKissR1 serum (1:1000), followed by detection using HRP-conjugated goat anti-rabbit IgG (Beyotime). Samples were probed in parallel with anti-tubulin antibody (Beyotime) as a control for protein loading.

Immunoreactive bands were detected with an enhanced chemiluminescent substrate (Beyotime), and the membrane was scanned using a Tanon 5200 Chemiluminescent Imaging System (Tanon Science and Technology, Shanghai, China).

## Ligand competition binding assay

A fluorescence-activated cell sorter (FACS) was used to detect the binding ability of kisspeptins with AjKissR1 or AjKissR2. HEK293 cells expressing FLAG–AjKissR1 or FLAG–AjKissR2 were washed with PBS that contained 0.2% bovine serum albumin (FACS buffer). We designed and synthesized an N-terminal FITC-labeled AjKiss1a peptide (see 'Key resources table'). Different kisspeptins were diluted in the FACS buffer to different concentrations, then added to cells that were incubated on ice for 60–90 min. Cells were washed thrice with the FACS buffer and re-suspended in the FACS buffer with 1% paraformaldehyde for 15 min. The binding activity of the indicated kisspeptin peptides with

AjKissR1 or AjKissR2 was determined by measuring the fluorescence of FITC and was presented as a percentage of total binding.

## Immunofluorescence assay on paraffin-embedded tissue sections

Paraffin sections were baked at 60°C for 2–4 hr and placed in xylene for 15 min, twice. The slides were washed twice in 100% ethanol for 10 min each, then in 95% ethanol for 10 min, 85% ethanol for 5 min, 70% ethanol for 5 min, and 50% ethanol for 5 min, followed by washing with $dH_2O$ for 5 min, and finally washing with PBS for 5 min. Antigen unmasking was performed in sodium citrate buffer (pH 6) for 10 min at 97°C, and then cooled to room temperature. Endogenous peroxidases were blocked by 10 min incubation in 3.0% hydrogen peroxide. Nonspecific antigens were blocked by a 60 min incubation in 0.3% bovine serum albumin (BSA) in TBST. Slides were incubated with primary antibodies overnight after removing the blocking solution, followed by 2 hr incubation with fluorescein isothiocyanate (FITC)-conjugated secondary antibodies (FITC-labeled goat anti-rabbit IgG (H+L), Beyotime). Slides were washed with $dH_2O$, mounted with antifade mounting medium (Beyotime), and imaged by confocal microscopy.

## Real-time quantitative PCR (qRT-PCR)

For qRT-PCR, *β-actin* (ACTB) and *β-tubulin* (TUBB) were chosen as the internal control (housekeeping) genes and gene-specific primers were designed based on the ORF sequences (*Xiang et al., 2016*; *Zhu et al., 2016*). Specific qRT-PCR primers for *AjKissR1/2* and *AjKiss1* were designed based on CDS (see 'Key resources table'). The primers were tested to ensure the amplification of single discrete bands, with no primer-dimers. qRT-PCR assays were carried out using the SYBR PrimeScript RT reagent Kit (TaKaRa, Kusatsu, Japan) following the manufacturer's instructions and using ABI 7500 Software v2.0.6 (Applied Biosystems, UK). The relative level of gene expression was calculated using the $2^{-\triangle Ct}$ method and data were normalized by geometric averaging of the internal control genes (*Livak and Schmittgen, 2001*; *Vandesompele et al., 2002*).

## Tissue culture and treatment

For in vitro experiments, the ovary and respiratory tree tissues were cut into small pieces of approximately 1 $mm^3$ and cultured in Leibovitz L-15 medium (HyClone) supplemented with 12.0 g/L NaCl, 0.32 g/L KCl, 0.36 g/L $CaCl_2$, 0.6 g/L $Na_2SO_4$, 2.4 g/L $MgCl_2$, 0.6 g/L glucose at 18°C in a humidified incubator (*Wang et al., 2020*). For ERK1/2 phosphorylation or immunofluorescence assays, samples were evaluated or fixed after 2 hr of ligand/PBS administration, with a 4 hr pre-treatment of pep234/PBS.

## Radioimmunoassay

Levels of estradiol (E2) in coelomic fluid or culture medium were measured using the Iodine ($^{125}$I) method (*Lu et al., 2016*). In brief, estradiol levels were measured using Iodine ($^{125}$I) radioimmunoassay kits (Beijing North Institute of Biotechnology, Beijing, China), according to the manufacturer's protocol. The binding rate is highly specific with an extremely low cross-activity to other naturally occurring steroids, which was less than 0.1% to most circulating steroids.

## Data statistics

Statistical analysis was done with GraphPad Prism (version 7.0). Statistical significance was determined using one-way ANOVA followed by Tukey's multiple comparisons test. Probability values that were less than or equal to 0.05 were considered significant, and less than or equal to 0.001 were considered extremely significant (*$p \leq 0.05$, **$p \leq 0.01$), and P values were indicated in the legend of the figures. All error bars represent the standard error of the mean (SEM), and all experimental data were gathered from at least three independent experiments showing similar results.

## Acknowledgements

The authors of this paper would like to thank Prof. Igor Yu Dolmatov from the National Scientific Center of Marine Biology—Russian Academy of Sciences for his assistance on histomorphological analysis, Dr Xiaoshang Ru from Institute of Oceanography, Chinese Academy of Sciences for his

assistance on sea cucumber sampling and suggestion on the discussion, Prof. Dongdong Xu for his technical assistance and equipment usage, and Dexiang Huang from Wang's lab for his technical assistance on phylogenetic analyses. This work was supported by the National Science Foundation of China (Nos. 41876154, 41406137 and 41606150) and Key Deployment Project of Center for Ocean Mega-Research of Science, Chinese Academy of Science (COMS2019Q15).

## Additional information

### Funding

| Funder | Grant reference number | Author |
|---|---|---|
| National Science Foundation of China | 41876154 | Tianming Wang |
| National Science Foundation of China | 41406137 | Tianming Wang |
| National Science Foundation of China | 41606150 | Jingwen Yang |
| Center for Ocean Mega-Research of Science, Chinese Acadamy of Science | COMS2019Q15 | Tianming Wang |

The funders had no role in study design, data collection and interpretation, or the decision to submit the work for publication.

### Author contributions

Tianming Wang, Conceptualization, Resources, Data curation, Formal analysis, Supervision, Funding acquisition, Validation, Visualization, Project administration; Zheng Cao, Zhangfei Shen, Data curation, Software, Formal analysis, Validation, Investigation, Visualization, Methodology; Jingwen Yang, Conceptualization, Resources, Supervision, Funding acquisition, Methodology, Project administration; Xu Chen, Validation, Investigation, Methodology; Zhen Yang, Ke Xu, Xiaowei Xiang, Qiuhan Yu, Yimin Song, Weiwei Wang, Yanan Tian, Investigation; Lina Sun, Formal analysis, Methodology; Libin Zhang, Funding acquisition, Investigation, Methodology; Su Guo, Conceptualization, Supervision, Validation, Project administration; Naiming Zhou, Conceptualization, Resources, Data curation, Software, Formal analysis, Supervision, Validation, Investigation, Visualization, Methodology, Project administration

### Author ORCIDs

Tianming Wang (iD) https://orcid.org/0000-0002-4499-4882
Naiming Zhou (iD) https://orcid.org/0000-0002-7571-9154

### Decision letter and Author response

Decision letter https://doi.org/10.7554/eLife.53370.sa1
Author response https://doi.org/10.7554/eLife.53370.sa2

## Additional files

### Supplementary files

• Transparent reporting form

### Data availability

All data generated or analysed during this study are included in the manuscript and supporting files. Source data files have been provided for Figures 1, 2, 3, 4, 5 and 7.

The following datasets were generated:

| | **Database and** |
|---|---|

| Author(s) | Year | Dataset title | Dataset URL | Identifier |
|---|---|---|---|---|
| Zhang X, Sun L, Yuan J, Sun Y, Gao Y, Zhang L, Li S, Dai H, Hamel J-F, Liu c, Yu Y, Liu S, Lin W, Guo K, Jin S, Xu P, Storey KB, Huan P, Zhang T, Zhou Y, Zhang J, Lin C, Li X, Xing L, Huo D, Sun M, Wang L, Mercier A, Li F, Yang h, Xiang J | 2017 | The sea cucumber genome provides insights into morphological evolution and visceral regeneration | https://www.ncbi.nlm.nih.gov/bioproject/PRJNA354676 | NCBI BioProject, PRJNA354676 |
| Tianming W | 2019 | Apostichopus japonicus kisspeptin receptor (Kissr1) mRNA, complete cds | http://www.ncbi.nlm.nih.gov/nuccore/MH709114 | NCBI GenBank, MH709114 |
| Wang T | 2019 | Apostichopus japonicus kisspeptin receptor (Kissr2) mRNA, complete cds | https://www.ncbi.nlm.nih.gov/nuccore/MH709115 | NCBI GenBank, MH709115 |

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
