## [Decision Letter]

**Acceptance summary:**

Your work is an impressive analysis of the kisspeptin signaling system ranging from the in silico identification, receptor deorphanization to functional tests in the sea cucumber. This is not only interesting in the context of the evolution of neuropeptidergic systems in bilaterians in general, but particularly also in understanding possible evolutionary ancient mechanisms underlying animal seasonality.

**Decision letter after peer review:**

Thank you for submitting your article "Existence and functions of hypothalamic kisspeptin neuropeptide signaling system in a non-chordate deuterostome species" for consideration by *eLife*. Your article has been reviewed by three peer reviewers, and the evaluation has been overseen by a Reviewing Editor and Diethard Tautz as the Senior Editor. The following individuals involved in review of your submission have agreed to reveal their identity: Maria Ina Arnone (Reviewer #2); Vera Terblanche (Reviewer #3).

The reviewers have discussed the reviews with one another and the Reviewing Editor has drafted this decision to help you prepare a revised submission.

Summary:

Wang et al. performed a detailed study of the echinoderm Kisspeptin/receptor system, using the sea cucumber *Apostichopus japonicus* as a model. They identified 2 preprohormones and three receptors, of which they study both peptide hormones and two receptors in detail. They performed phylogenetic analyses, receptor activation and signaling studies, Western blot analyses and immunohistochemistry. The authors also provide an initial characterization of this peptidergic system in *Apostichopus japonicus* physiology. Based on those studies they come to important conclusions about the evolutionary conservation of the kisspeptin system.

Their findings are of potential great interest, being the first demonstration of functioning of this system outside vertebrates, with important implications for the understanding of the evolution of vertebrate (neuro)secretory systems, as well as seasonal regulation of reproduction and physiology.

However, the manuscript requires several essential improvements especially in its presentation (details below).

Essential revisions:

1) The technical information necessary to evaluate the manuscript is either missing or spread across various places. This includes all primary metadata, sequences of synthetic peptides used for receptor, physiological and antibody studies. Metadata are entirely missing. In case of the peptides, it is not specified in the main text whether the synthetic Kiss peptides used in the experiment are amidated. This could only be found when digging through the supplementary tables.

2) Connected to the missing technical information, there is insufficient information on the replicates used in the different studies. For instance, regarding the Ca^2+^ mobilization experiments (Figure 2B1 and B2). How representative are these experiments? How many times were they repeated and how reproducible were the results? In other cases a number "n" for replicates is mentioned, but it is unclear how the replicates (e.g., technical, biological) were defined.

3) The phylogenetic trees require major improvements. De-code accession numbers in trees for species to make it better understandable to the reader without having to consult the databases for every branch. This is essential to see what the tree is really showing (and how representative it is.)

The critical bootstrap value for the Kisspeptin family preprohormone group is only 34 – this means there is no real support for this group as a common branch and hence, in a strict sense, this would argue against a common evolutionary ancestry. On the other hand- such seemingly unsupported grouping is not totally uncommon for preprohormones across larger phylogenetic distances (e.g. due to the relative shortness of the peptide chains that can be used for the comparison), but then the authors need to discuss this issue and provide other reasons (such as their receptor homology, deorphanization and cross-binding studies) to argue their case for a preprohormone group that derived from a common evolutionary ancestor. Genomic gene structure information on the pre-prohormone could provide additional support (e.g. exon/intron boundaries; microsyntheny).

4) Figure 5D, E: The data on EGFP-PKC isoform recruitment to the plasma membrane assays shown in Figure 5D are not convincing. In some cases the potential membrane signal is largely obscured by the high cytoplasmic expression. Is there a way to quantify this using a histogram? Maybe this figure is also not strictly necessary to support the findings.

5) The specificity of the antibodies raised against AjKiss1b-10 and AjKISS-R1 is of major concerns.

There should be experiments showing that the antibody recognizes specifically the antigen. For protein blots, this could be achieved by probing lysates of non-transfected and transfected mammalian neuroendocrine cells, which could process the AjKiss precursor. The same goes for immunofluorescence, e.g. compare transfected and plasmid-only transfected tissue culture cells. In AjKISS-R1 transfected cells does the antibody signal co-localize with the GFP signal to the membrane?

Protein blots:

What kind of gel did the authors use for this experiment in Figure 6A? AjKiss1b-10 is 10 amino acid long (~ 1.1 kDa). Since peptides this small are unlikely to be retained after transfer to PVDF, the authors should specify the experimental procedure to obtain this blot. Molecular weight markers would be useful to confirm the signals from the Aj tissues sampled are from the unprocessed or the partially processed precursor or the mature peptide. Was the anti-AjKiss1b-10 raised against the amidated peptide? If so, why the signals from the mature peptide are not detected ("Western Blot analysis of *A. japonicus* kisspeptin precursor")? Why is Kisspeptin detected in OVA by western and not by immunofluorescence?

6) There are further major omissions of explanations throughout the text, e.g.:

In Figure 4A, the authors completely omit to discuss the PTX treatment shown in the panels. Does PTX stand for Pertussis toxin? The authors need to mention this treatment in the text and explain the reason to use PTX.

Explain AjKiss1b-10 in the text. Same for in the Figure 4 legend. Please consider that not all readers are familiar with the literature on Kisspeptins or neuropeptides in general. Also, please, pick one naming system and keep using it; i.e., use either Kp or Kiss or Kisspeptin. Same for receptor naming.

Subsection “In silico identification of Kps and Kp receptors”, first paragraph: is that the same sequence identified by Swansa-ard et al./Chen et al? If yes state that and those lines can be omitted as it also repeats the Introduction.

Figure 5A1. What time (5 min, 10 min, etc. stimulation) the concentration-dependence blots refer to?

Figure 5A2 (bar graph). Why isn't the strongest signal for AjKissR2 set at 100% as in the case of AjKissR1?

Figure 6D. Upper left panel. What treatment is this?

Figure 6F. Explain what tissue index measures.

Figure 7: Why does this figure indicate presence of only one kisspeptin receptor in sea cucumber if the presented study finds two?

Substantial editing work is needed to allow readers to more easily follow the data. Generally, the figure legends and the Materials and methods section need the most attention.

Figures. Make larger panels, please. Some of the labels on the panels were very difficult, if not impossible, to decipher (for instance, 1C, 1E, 5A, 5B, 5C). Check for typos in the figure labels (for instance, Figure 2B: 1 μM instead of 1 um).

Materials and methods.

Animals should not be listed as "Materials". Transient and stable HEK293 cells expressing AjKissRs were generated, but it is not explained why. A protocol for DiI staining (Figure 2A) is not provided.

Receptor internalization (shown in Figure 2C) mentions 30 min, but the legend says 60 min. In the same experiments, it is said that cells were incubated with DAPI for several minutes before translocation. DAPI is not a vital stain; did the authors mean Hoechst? Species name: sometimes *A. japonicas*, at others it is *A. japonicus.*

Subsection “Data statistics”: please refer to the exact experiments for which these statistic methods were used.

7) In general, a conclusive sentence about what each of the experiments shows would greatly help the reader, e.g.- Subsection “Physiological functions of the Kp signaling system in *A. japonicus*”, third paragraph: a one line conclusion on kiss signalling functions inferred from the effects of persistent kiss1a/1b treatment would be nice.

Do you think it maintains metabolic balance, or does it lead to the mobilisation of lipids from the fat tissues?

8) The data on the potential function in seasonal control of reproduction are interesting, but as presented at present rather confusing, as they seem to be partly somewhat contradictory.

The statements in the second and last paragraphs of the subsection “Physiological functions of the Kp signaling system in *A. japonicus*, would suggest an important role in regulating the maturation of the germline, especially oocytes before/during the peak season of reproduction. How does this fit with the findings described in the third paragraph of the aforementioned subsection, describing what rather seems to be a role in aestivation?

9) Final conclusion: The statement that it "originally evolved in this hypothalamic neuropeptide system" implies that it evolved along with the hypothalamus in higher vertebrates. The work presented by the authors does not support this claim, but rather implies that the molecular system itself (including the use of Gaq-PKC cascade in Kp signalling) is likely an ancestral state and is now used by the vertebrate hypothalamus. To draw evolutionary conclusion about an ancestral hypothalamic system the study would have to include other markers for vertebrate hypothalamic cell types (e.g. nkx2.1) and compare how kisspeptin localizes relative to the expression of these other markers. Furthermore, some places of kisspeptin detection- OVA, Tes, ANP- don't make sense with an "ancestral neurosecretory system" in the sense of an ancestral tissue unit. The authors should revise their text, heading and conclusions according to this, rather pointing out the molecular conservation, as well as the possibly interesting conserved link in the regulation of seasonal reproduction.

[Editors' note: further revisions were suggested prior to acceptance, as described below.]

Thank you for resubmitting your work entitled "Existence and functions of hypothalamic kisspeptin neuropeptide signaling system in a non-chordate deuterostome species" for further consideration by *eLife*. Your revised article has been reviewed by three peer reviewers, one of whom is a member of our Board of Reviewing Editors, and the evaluation has been overseen by Diethard Tautz as the Senior Editor.

The manuscript has been significantly improved by the additional experiments and clarifications, but there are some remaining issues that need to be addressed before acceptance, as outlined below. All aspects can be addressed by text or figure amendments. There are no further experiments needed.

1) At present there is no strong evolutionary evidence for this system to homologous to the hypothalamus. We would thus ask to remove the word "hypothalamic" from the title.

Title suggestion:

"Existence and functions of a kisspeptin neuropeptide signaling system in a non-chordate deuterostome species"

2) "Through the evaluation of Ca^2+^ mobilization and other intracellular signals, we found that *A. japonicus* kisspeptins dramatically activated two receptors (AjKissR1 and AjKissR2), via a GPCR-mediated Gαq/PLC/PKC/MAPK signaling pathway, that have functions corresponding to those of the vertebrate kisspeptin system". There is something wrong with this statement. Again, the GPCR-mediated Gαq/PLC/PKC/MAPK signaling pathway was shown in a mammalian cell line surrogate. So, while ex vivo experiments show ERK (i.e., MAPK cascade) involvement (Figure 7C and D) the authors have not shown that in/ex vivo Aj kisspeptin signaling involves PLC and PKC.

We would thus suggest to rephrase this in the following way (or something similar):

"Through the evaluation of Ca^2+^ mobilization and other intracellular signals, we found that *A. japonicus* kisspeptins activate two receptors (AjKissR1 and AjKissR2) via a GPCR-mediated Gαq/PLC/PKC/MAPK signaling pathway in a mammalian cell line. Albeit likely it remains to be shown if the same signaling cascade also occurs in vivo in its seemingly conserved function in reproductive control."

3) Subsection “In silico identification of kisspeptins and kisspeptin receptors”. Change "ANP, containing nerve ring" to "ANP, containing the nerve ring".

4) Subsection “In silico identification of kisspeptins and kisspeptin receptors”. Change "frames" to "frame".

5) Figure 3B legend. Change "3-B" to "3B".

6) Figure 4 legend. It recalls the Figure 4 source data. Here, there is a mislabelling, as the linked excel files has individual worksheet labelled as 3A, 3B etc. – they should be 4A, 4B etc, instead. The authors should check all their supplementary files for correct labelling.

7) "inhibitory effect of pep234 was preapproved in vitro", should be changed to "inhibitory effect of pep234 was validated in vitro".

8) "Further detection of the pERK signal in AjKiss1b-10 treated oocytes by confocal microscopy demonstrated the physiological activation of this pathway by AjKiss1b-10 and pep234 on *A. japonicus* cells" should be changed to "Further, detection by confocal microscopy of the p-ERK signal in treated oocytes demonstrated activation of this pathway by AjKiss1b-10 and its inhibition by pep234 in *A. japonicus* cells".

9) "Samples were collected and fixed after 2 h of ligand administration with or without a 4 h pre-treatment of pep234, in optimized L15 medium at 18 °C". Should be changed to "Samples were collected and fixed after 2 h of ligand administration with or without a 4 h pre-treatment with pep234, in optimized L15 medium at 18 °C". Also, is this the same treatment for the samples shown in Figure 7C? If so, it should be clearly stated in the legend for panel 7C.

10) After "different potency", we suggest the addition of ", when expressed in a mammalian cell line".

11) Change statement from "therefore proving the functionality of this neuropeptide system in non-chordate species" to "therefore proving that this peptide system is present and active in non-chordate deuterostome species".

12) "an amphioxus kisspeptin receptor was shown to trigger significant PKC and not PKA signaling, when stimulated by two kisspeptin-type peptides (Wang et al., 2017)." Please, make clear that these experiments too were done using heterologous expression in HEK293 cells.

13) It is stated that "AjKissR1 and AjKissR2 induced ERK1/2 activation via a Gαq/PLC/PKC cascade. Our results showed that no significant accumulation of cAMP was detected in response to agonist treatment". We could not find any experiment in the manuscript measuring cAMP accumulation. The authors should amend the text accordingly.

14) All images of the Western blots are over-contrasted. This needs to be corrected and size markers also added to Figure 7 (not just the figure supplement).

15) Carefully check for missing and correct scalebars. Figure 3: According to the scale bar these HEK cells are huge. Sure the scale bar is correct? Figure 6D, E: scale bars are entirely missing.

16) There is still some remaining concern about the specificity of the antibody against AjKiss1b. We find it pretty unusual that an antibody that was generated against an amidated peptide would now only rather recognize the precursor (or if we played devil's advocate- a much larger band in the Western that could have nothing to do with the actual peptide). We would thus ask the authors to insert a cautionary sentence in the text.

---

## [Author Response]

Essential revisions:1) The technical information necessary to evaluate the manuscript is either missing or spread across various places. This includes all primary metadata, sequences of synthetic peptides used for receptor, physiological and antibody studies. Metadata are entirely missing. In case of the peptides, it is not specified in the main text whether the synthetic Kiss peptides used in the experiment are amidated. This could only be found when digging through the supplementary tables.

As the reviewers suggested, in the revised manuscript, we have added the primary metadata in xlsx or txt format linked with the figures in main text or the supplementary figures, and the Key Resources Table is now incorporated within the main text at the very beginning of the Materials and methods section to provide the detailed information about the sequences and amidation of indicated synthetic peptides, as well as the characters of antibodies, especially for the customized polyclonal antibody for AjKiss1b-10 and AjKissR1. We thank the reviewers for their suggestion.

2) Connected to the missing technical information, there is insufficient information on the replicates used in the different studies. For instance, regarding the Ca^2+^ mobilization experiments (Figure 2B1 and B2). How representative are these experiments? How many times were they repeated and how reproducible were the results? In other cases a number "n" for replicates is mentioned, but it is unclear how the replicates (e.g., technical, biological) were defined.

We have now provided the information on the replicates used in different experiments, which is now introduced in every figure legend and the “n” for animals or sample numbers is also clearly denoted in the revised manuscript as the reviewers suggested, and we thank the reviewers for this suggestion.

3) The phylogenetic trees require major improvements. De-code accession numbers in trees for species to make it better understandable to the reader without having to consult the databases for every branch. This is essential to see what the tree is really showing (and how representative it is.)The critical bootstrap value for the Kisspeptin family preprohormone group is only 34 – this means there is no real support for this group as a common branch and hence, in a strict sense, this would argue against a common evolutionary ancestry. On the other hand- such seemingly unsupported grouping is not totally uncommon for preprohormones across larger phylogenetic distances (e.g. due to the relative shortness of the peptide chains that can be used for the comparison), but then the authors need to discuss this issue and provide other reasons (such as their receptor homology, deorphanization and cross-bindind studies) to argue their case for a preprohormone group that derived from a common evolutionary ancestor. Genomic gene structure information on the pre-prohormone could provide additional support (e.g. exon/intron boundaries; microsyntheny).

We thank the reviewers for pointing out these issues. The abbreviated Latin names of species and protein names are now represented in trees, instead of accession numbers, to make it better understandable to the readers. To get more representative trees for preprohormone groups, a new strategy for ML tree construction is now applied in kisspeptin and kisspeptin receptor phylogenetic analyses. As introduced in the Materials and methods section of the revised text, our new approach mainly includes using Constraint-based Multiple Alignment Tool (https://www.ncbi.nlm.nih.gov/tools/cobalt/cobalt.cgi?CMD=Web) for the multiple protein sequence alignment, and the FastTree program (version 2.1) for construction of the trees based on WAG and CAT model, and providing local support values based on the Shimodaira-Hasegawa (SH) test. For the phylogenetic tree of peptides, the poorly aligned regions from the multiple sequence alignment are automatically removed by trimAl. The trees were then visualized through Evolview web server. The revised phylogenetic tree for kisspeptin analysis represents much higher local support value for topological stability of the tree. Though, in the current tree, the AjKiss is clustered with predicated kisspeptin from other two sea cucumbers and then grouped with chordate kisspeptins under the high local support value of 81%, we still believe that the bioinformatical analysis can’t support the definition of this screened Kisspeptin-like gene. We do agree with reviewers that gene structure information on the pre-prohormones could provide additional support and now it has been provided in the revised Figure 1. The conserved gene structure is displayed from comparative analysis of sea cucumber, zebrafish and human *preKiss* genes. We believe that these data, combined with the receptor homology, deorphanization and cross-binding results from our studies, should be sufficiently convincing to demonstrate the evolutionary homology of AjKiss and chordate kisspeptins.

4) Figure 5D, E: The data on EGFP-PKC isoform recruitment to the plasma membrane assays shown in Figure 5D are not convincing. In some cases the potential membrane signal is largely obscured by the high cytoplasmic expression. Is there a way to quantify this using a histogram? Maybe this figure is also not strictly necessary to support the findings.

As the reviewers pointed out, we have performed additional experiments to address the concern on PKC isoform recruitment to the plasma membrane assays using plasmids of GFP-tagged rat PKC isoforms (α, βI, βII and δ) (details are shown in Key Resources Table) and obtained much more convincing data. As shown in the now revised Figure 6D and E, upon activation by AjKiss^-1^b, PKCα, βI and βII were clearly translocated to the plasma membrane, but not PKCδ. These new data suggest that PKCα, βI and βII are involved in the AjKissR1 and R2-mediated downstream signaling pathways.

5) The specificity of the antibodies raised against AjKiss1b-10 and AjKISS-R1 is of major concerns.There should be experiments showing that the antibody recognizes specifically the antigen. For protein blots, this could be achieved by probing lysates of non-transfected and transfected mammalian neuroendocrine cells, which could process the AjKiss precursor. The same goes for immunofluorescence, e.g. compare transfected and plasmid-only transfected tissue culture cells. In AjKISS-R1 transfected cells does the antibody signal co-localize with the GFP signal to the membrane?

Detection of kisspeptin in non-transfected vs. transfected mammalian neuroendocrine cells should be a good approach for the analysis of intracellular processing and for assessing specificity of the antibody. We tried to develop this expression system using transfected mammalian neuroendocrine cells but failed due to their slow growth hence very difficult to culture. In addition, we were not allowed to enter the laboratory until now due to the coronavirus outbreak.

Therefore, to provide conclusive evidence of the antibody specificity raised against AjKiss1b-10, instead, we have collected fresh tissue samples from the sea cucumber for western blotting. As shown in Figure 7—figure supplement 1A, one band of AjKiss1b-1 (or AjKiss1b) and another about 20 KDa band can be detected. This 20 KDa band in tissue samples should be *Apostichopus japinicus* Kisspeptin precursor (theoretically deduced molecular weight of AjpreKiss is 20.29 KDa).

For assessing the specificity of anti- AjKissR1 anti-body, we performed western blot and immunofluorescence assays in HEK293 cells. As shown in Figure 7—figure supplement 1B and C, using anti- AjKissR1 antibody by western blot, we detected a 65 KDa band of AjKissR1-EGFP (theoretically deduced molecular weight 68 KDa) from HEK 293 cells; we also showed a 45 KDa band from the sea cucumber ovarian sample (theoretically deduced molecular weight 43.46 KDa). Using the immunofluorescence assay, co-localization of Cy3 and GFP in AjKissR1-expressing cells, but not in AjKissR2-expressing cells, was observed by confocal microscopy. Collectively, these data have demonstrated that the antibodies raised against AjKiss1b-10 and AjKISS-R1 are specifically recognizing the targeted proteins.

Protein blots:What kind of gel did the authors use for this experiment in Figure 6A? AjKiss1b-10 is 10 amino acid long (~ 1.1 kDa). Since peptides this small are unlikely to be retained after transfer to PVDF, the authors should specify the experimental procedure to obtain this blot. Molecular weight markers would be useful to confirm the signals from the Aj tissues sampled are from the unprocessed or the partially processed precursor or the mature peptide. Was the anti-AjKiss1b-10 raised against the amidated peptide? If so, why the signals from the mature peptide are not detected ("Western Blot analysis of *A. japonicus* kisspeptin precursor")? Why is Kisspeptin detected in OVA by western and not by immunofluorescence?

To detect kisspeptin precursor or its mature peptide in different tissues of sea cucumber, protein samples were electrophoresed on a 15% SDS polyacrylamide gel (details are provided in the Materials and methods section of revised text) and transferred to PVDF membranes (180 mA, 40 min for kisspeptin detection). Molecular weight markers imaged in bright field are now provided to show the size of the bands (as shown in Figure 7—figure supplement 1). The results indicate that the detected signal mainly shows the unprocessed precursor.

The anti-AjKiss1b-10 antibody was raised against the amidated peptide, but the signals from the mature peptide are not detected, because the concentration of the mature peptide in tissue samples is too low to be detected; it may also be possible that this antibody does not recognize the CONH_2_ group.

In the current study, we detected kisspeptin precursor in OVA by western, but not by immunofluorescence. This may be due to the specific structures and components of oocytes, as well as the characteristics of the antibody. The presence of the ligand in widely distributed secretory vesicles, which is usually involved in kisspeptin processing and transporting (Hiroko Murakawa, et al., 2016), may be difficult to detect in big oocytes (about 150-200 μm in diameter) by confocal microscopy. For western blotting, the lipids and associated proteins (main components) in mature oocytes are eliminated during sample preparation, consequently increases the concentration of kisspeptin, making it detectable by western blot. Alternatively, it may be also due to the characteristics of the antibody. In fact, many commercially available antibodies are only good for western, not for histochemistry. We explained this inconsistence in revised main text as “the inconsistency with results from the western blotting assay vs. immunofluorescence may be either due to specific structures and components of oocytes, or the characteristics of the antibody”.

Murakawa, H., Iwata, K., Takeshita, T., and Ozawa, H. (2016). Immunoelectron microscopic observation of the subcellular localization of kisspeptin, neurokinin B and dynorphin A in KNDy neurons in the arcuate nucleus of the female rat.Neuroscience Letters, 612, 161-166. doi: 10.1016/j.neulet.2015.12.008

6) There are further major omissions of explanations throughout the text, e.g.:In Figure 4A, the authors completely omit to discuss the PTX treatment shown in the panels. Does PTX stand for Pertussis toxin? The authors need to mention this treatment in the text and explain the reason to use PTX.

We apologize for this omission. PTX does stand for Pertussis toxin. We used PTX to exclude the involvement of Gi protein in the AjKiss receptors-mediated signaling. We have amended the text accordingly.

Explain AjKiss1b-10 in the text. Same for in the Figure 4 legend. Please consider that not all readers are familiar with the literature on Kisspeptins or neuropeptides in general. Also, please, pick one naming system and keep using it; i.e., use either Kp or Kiss or Kisspeptin. Same for receptor naming.

As the reviewers suggested, we have listed the detailed information of AjKiss1b-10 and pep234, as well as other peptides mentioned in this article in Key Resources Table, and also unified usage of Kiss and KissR in the revised version of manuscript.

Subsection “In silico identification of Kps and Kp receptors”, first paragraph: is that the same sequence identified by Swansa-ard et al/Chen et al? If yes state that and those lines can be omitted as it also repeats the Introduction.

The lines have been removed in the revised manuscript as the reviewers suggested.

Figure 5A1. What time (5 min, 10 min, etc. stimulation) the concentration-dependence blots refer to?

We challenged the cells with different concentrations of ligand for 5 min in the concentration-dependence blots, and we have provided this info in the revised manuscript.

Figure 5A2 (bar graph). Why isn't the strongest signal for AjKissR2 set at 100% as in the case of AjKissR1?

As shown in the now revised Figure 6A1, AjKissR1 exhibited higher efficacy and potency than that of AjKissR2, therefore, when we deal with the data of Figure 5A2 (bar graph), the ratio of p-ERK1/2/total ERK1/2 was normalized to peak value detected in corresponding experiments and the strongest signal of AjKissR1 was set as 100%, so that we can easily compare the ERK1/2 activity mediated by AjKissR1 and AjKissR2.

Figure 6D. Upper left panel. What treatment is this?

As the reviewers pointed out, we have clarified the treatment for upper left panel of the now revised Figure 7D.

Figure 6F. Explain what tissue index measures.

In the now revised Figure 7F, “tissue index” refers to the ratio of tissue weight/body weight. The detailed information of calculation, as well as the statistics, is listed in metadata.

Figure 7: Why does this figure indicate presence of only one kisspeptin receptor in sea cucumber if the presented study finds two?

We apologize for the omission. We have redrawn this figure (Figure 8 in the revised version) to provide more accurate information about the current discovery about the kiss system.

Substantial editing work is needed to allow readers to more easily follow the data. Generally, the figure legends and the Materials and methods section need the most attention.

As the reviewers suggested, we have carefully gone through the whole manuscript and corrected typos and grammatical errors, especially in the figure legends and the Materials and methods section.

Figures. Make larger panels, please. Some of the labels on the panels were very difficult, if not impossible, to decipher (for instance, 1C, 1E, 5A, 5B, 5C). Check for typos in the Figure labels (for instance, Figure 2B: 1 μM instead of 1 um).

We have enlarged figure panels together with the label, and also corrected typos in the figure label as the reviewers suggested.

Materials and methods.Animals should not be listed as "Materials". Transient and stable HEK293 cells expressing AjKissRs were generated, but it is not explained why. A protocol for DiI staining (Figure 2A) is not provided.

As the reviewers suggested, we have added the animals-related content to the “Animal collection and treatment” subtitle following the Key Resources Table in Materials and methods section. Establishment of stable expression cell lines can reduce the influence caused by different transfection efficiency. We have also added description of the DiI staining protocol in the Materials and methods section.

Receptor internalization (shown in Figure 2C) mentions 30 min, but the legend says 60 min. In the same experiments, it is said that cells were incubated with DAPI for several minutes before translocation. DAPI is not a vital stain; did the authors mean Hoechst? Species name: sometimes *A. japonicas*, at others it is *A. japonicus*.

We have to apologize for our carelessness in preparing our manuscript, and we have provided more detailed information about DiI and DAPI treatment in the Materials and methods section of revised manuscript as the reviewers suggested. By the way, just like Hoechst, DAPI is a popular nuclear and chromosome counterstain, and emits blue fluorescence upon binding to AT regions of DNA (Biotech Histochem.1995;70(5):220-33; Cell2006;125(4):679-90). Additionally, we have also unified the usage of “*A. japonicus*” in the whole revised manuscript. We thank the reviewers for these suggestions.

Subsection “Data statistics”: please refer to the exact experiments for which these statistic methods were used.

As the reviewers suggested, we have added the description on data statistics in the legend of the exact experiments for which these statistic methods were used in the revised manuscript, and thank the reviewers for this suggestion.

7) In general, a conclusive sentence about what each of the experiments shows would greatly help the reader, e.g.- Subsection “Physiological functions of the Kp signaling system in *A. japonicus*”, third paragraph: a one line conclusion on kiss signalling functions inferred from the effects of persistent kiss1a/1b treatment would be nice.Do you think it maintains metabolic balance, or does it lead to the mobilisation of lipids from the fat tissues?

As the reviewers constructively suggested, we have added a conclusive sentence “These data suggest that the *A. japonicus* kisspeptin system plays a role in the control of metabolic balance”. As for the exact physiological role of the *A. japonicus* kisspeptin system, e.g. mobilisation of lipids from the fat tissues, it is in our future research plan.

8) The data on the potential function in seasonal control of reproduction are interesting, but as presented at present rather confusing, as they seem to be partly somewhat contradictory.The statements in the second and last paragraphs of the subsection “Physiological functions of the Kp signaling system in *A. japonicus*, would suggest an important role in regulating the maturation of the germline, especially oocytes before/during the peak season of reproduction. How does this fit with the findings described in the third paragraph of the aforementioned subsection, describing what rather seems to be a role in aestivation?

As the reviewers pointed out, in the subsection “Physiological functions of the kisspeptin signaling system in *A. japonicus*”, we intended to describe the role of kiss system in the metabolic control of sea cucumber. In sea cucumber, aestivation is directly related to metabolism and reproduction, and PK expression and activity is an important physiological index for metabolism and reproduction. Although we don’t have strong evidence to support the functional role of kisspeptin in aestivation of sea cucumber, our data indicate the important role of kisspeptin in the regulation of reproductive development and the metabolic balance of sea cucumber. These two different physiological processes arise simultaneously in the seasonal reproduction of *Apostichopus japonicus* (Ru et al., 2017 and 2018). Moreover, the latest data from Ru’s research (data unpublished, paper under revision) indicate that reproduction causes a significant increase in oxygen consumption in *A. japonicus*, and sea cucumber can accommodate the high oxygen demand by accelerating respiratory rate (this is consistent with the elevation of *PK* RNA expression level of respiratory tree under kisspeptin administration). Meanwhile, Ru’s research also found the decreased appetite during reproductive development which should be related with the degeneration of intestine. Collectively, we believe that kisspeptin administration induces both metabolic regulation (expressional elevation of *PK* and degeneration of intestine) and reproductive control (expression of kisspeptin and its receptor in germline).

Ru, X., Zhang, L., Liu, S., and Yang, H. (2017). Reproduction affects locomotor behaviour and muscle physiology in the sea cucumber, *Apostichopus japonicus*. Animal Behaviour, 133, 223–228. doi:10.1016/j.anbehav.2017.09.024

Ru, X., Zhang, L., Liu, S., Sun, J., and Yang, H. (2018). Energy budget adjustment of sea cucumber *Apostichopus japonicus* during breeding period. Aquaculture Research, 49(4), 1657–1663. doi:10.1111/are.13621

Unpublished paper: Plasticity of respiratory function accommodates high oxygen demand in breeding sea cucumbers

9) Final conclusion: The statement that it "originally evolved in this hypothalamic neuropeptide system" implies that it evolved along with the hypothalamus in higher vertebrates. The work presented by the authors does not support this claim, but rather implies that the molecular system itself (including the use of Gaq-PKC cascade in Kp signalling) is likely an ancestral state and is now used by the vertebrate hypothalamus. To draw evolutionary conclusion about an ancestral hypothalamic system the study would have to include other markers for vertebrate hypothalamic cell types (e.g. nkx2.1) and compare how kisspeptin localizes relative to the expression of these other markers. Furthermore, some places of kisspeptin detection- OVA, Tes, ANP- don't make sense with an "ancestral neurosecretory system" in the sense of an ancestral tissue unit. The authors should revise their text, heading and conclusions according to this, rather pointing out the molecular conservation, as well as the possibly interesting conserved link in the regulation of seasonal reproduction.

As the reviewers suggested, we have removed “originally evolved in this hypothalamic neuropeptide system”, and added one sentence “G_αq_-coupled signaling is highly conserved in the kisspeptin signaling systems from *A. japonicus* to mammals”. We have already revised corresponding text, heading and conclusions according to reviewers’ suggestion. We thank the reviewers for this constructive suggestion.

[Editors' note: further revisions were suggested prior to acceptance, as described below.]The manuscript has been significantly improved by the additional experiments and clarifications, but there are some remaining issues that need to be addressed before acceptance, as outlined below. All aspects can be addressed by text or figure amendments. There are no further experiments needed.1) At present there is no strong evolutionary evidence for this system to homologous to the hypothalamus. We would thus ask to remove the word "hypothalamic" from the title.Title suggestion:"Existence and functions of a kisspeptin neuropeptide signaling system in a non-chordate deuterostome species"

We replaced the word “hypothalamic” with “a” in the title. We thank the reviewers for this suggestion.

2) "Through the evaluation of Ca^2+^ mobilization and other intracellular signals, we found that *A. japonicus* kisspeptins dramatically activated two receptors (AjKissR1 and AjKissR2), via a GPCR-mediated Gαq/PLC/PKC/MAPK signaling pathway, that have functions corresponding to those of the vertebrate kisspeptin system". There is something wrong with this statement. Again, the GPCR-mediated Gαq/PLC/PKC/MAPK signaling pathway was shown in a mammalian cell line surrogate. So, while ex vivo experiments show ERK (i.e., MAPK cascade) involvement (Figure 7C and D) the authors have not shown that in/ex vivo Aj kisspeptin signaling involves PLC and PKC.We would thus suggest to rephrase this in the following way (or something similar):"Through the evaluation of Ca^2+^ mobilization and other intracellular signals, we found that *A. japonicus* kisspeptins activate two receptors (AjKissR1 and AjKissR2) via a GPCR-mediated Gαq/PLC/PKC/MAPK signaling pathway in a mammalian cell line. Albeit likely it remains to be shown if the same signaling cascade also occurs in vivo in its seemingly conserved function in reproductive control."

As the reviewers suggested, we have rephrased these sentences in the revised version. We thank the reviewers for this suggestion.

3) Subsection “In silico identification of kisspeptins and kisspeptin receptors”. Change "ANP, containing nerve ring" to "ANP, containing the nerve ring".

We have added “the” to this sentence. We thank the reviewers.

4) Subsection “In silico identification of kisspeptins and kisspeptin receptors”. Change "frames" to "frame".

We have corrected it in revision and thank the reviewers.

5) Figure 3B legend. Change "3-B" to "3B".

The “-” has been deleted. We thank the reviewers.

6) Figure 4 legend. It recalls the Figure 4 source data. Here, there is a mislabelling, as the linked excel files has individual worksheet labelled as 3A, 3B etc. – they should be 4A, 4B etc, instead. The authors should check all their supplementary files for correct labelling.

We have corrected these labels and checked all the labels in our supplementary files. Besides, an error of data in the source data sheet of “Figure 7H FEB” (cells C27-C29) was found and the source data file (Figure 7E+F+H metadata), as well as the heatmap in Figure 7, was corrected accordingly in revision. We apologize for the carelessness in writing and data organization. We thank the reviewers.

7) "inhibitory effect of pep234 was preapproved in vitro", should be changed to "inhibitory effect of pep234 was validated in vitro".

The sentence has been optimized. We thank the reviewers.

8) "Further detection of the pERK signal in AjKiss1b-10 treated oocytes by confocal microscopy demonstrated the physiological activation of this pathway by AjKiss1b-10 and pep234 on *A. japonicus* cells" should be changed to "Further, detection by confocal microscopy of the p-ERK signal in treated oocytes demonstrated activation of this pathway by AjKiss1b-10 and its inhibition by pep234 in *A. japonicus* cells".

We have rewritten the sentence following the reviewers’ comments. We thank the reviewers.

9) "Samples were collected and fixed after 2 h of ligand administration with or without a 4 h pre-treatment of pep234, in optimized L15 medium at 18 °C". Should be changed to "Samples were collected and fixed after 2 h of ligand administration with or without a 4 h pre-treatment with pep234, in optimized L15 medium at 18 °C". Also, is this the same treatment for the samples shown in Figure 7C? If so, it should be clearly stated in the legend for panel 7C.

The legend has been revised following reviewers’ comments. We thank the reviewers.

10) After "different potency", we suggest the addition of ", when expressed in a mammalian cell line".

The sentence has been revised following reviewers’ comments. We thank the reviewers.

11) Change statement from "therefore proving the functionality of this neuropeptide system in non-chordate species" to "therefore proving that this peptide system is present and active in non-chordate deuterostome species".

The sentence has been revised following reviewers’ comments. We thank the reviewers.

12) "an amphioxus kisspeptin receptor was shown to trigger significant PKC and not PKA signaling, when stimulated by two kisspeptin-type peptides (Wang et al., 2017)." Please, make clear that these experiments too were done using heterologous expression in HEK293 cells.

An additional description “using heterologous expression in cultured HEK293 cells” was added to this sentence in text. We thank the reviewers.

13) It is stated that "AjKissR1 and AjKissR2 induced ERK1/2 activation via a Gαq/PLC/PKC cascade. Our results showed that no significant accumulation of cAMP was detected in response to agonist treatment". We could not find any experiment in the manuscript measuring cAMP accumulation. The authors should amend the text accordingly.

This statement has been removed and an additional sentence “we failed to collect distinct evidence to prove the involvement of G_αs_-dependent signaling in *A. japonicus* Kisspeptin system.” was added following “Although the G_αs_ protein has been shown to be implicated in the teleost kisspeptin receptors-mediated signaling pathway,”. These sentences were amended accordingly. We thank the reviewers.

14) All images of the Western blots are over-contrasted. This needs to be corrected and size markers also added to Figure 7 (not just the figure supplement).

We have checked and revised all images of the Western blots as the reviewers suggested. Besides, we found that the WB images in Figure 7—figure supplement 3B, for the inhibitory activity of pep234 in AjKissR2 mediated ERK signaling, was upside down in the previous version. We have also corrected the data in revision. We thank the reviewers for pointing out this issue.

15) Carefully check for missing and correct scalebars. Figure 3: According to the scale bar these HEK cells are huge. Sure the scale bar is correct? Figure 6D, E: scale bars are entirely missing.

We apologize for the incorrect and missing scalebars. We have checked all the scalebars referring to primary data. The “20 μm” in Figure 3 was marked by mistake, and it has been revised as “10 μm”. The scalebars in Figure 6 have been added. Meanwhile, the scale bars in Figure 6—figure supplement 1 were adjusted too. We thank the reviewers for pointing out this issue.

16) There is still some remaining concern about the specificity of the antibody against AjKiss1b. We find it pretty unusual that an antibody that was generated against an amidated peptide would now only rather recognize the precursor (or if we played devil's advocate- a much larger band in the Western that could have nothing to do with the actual peptide). We would thus ask the authors to insert a cautionary sentence in the text.

Thank the reviewers for this comment. We have added a cautionary sentence, “However, the failure to detect mature peptide fragment using anti-AjKiss1b-10 indicates that the further development of antibodies with highsensitivityandspecificity might be required to clarify the mature kisspeptin location in *A. japonicus* tissues. ” following suggestion.